

# Diazotrophy as a key driver of the response of marine net primary productivity to climate change

Laurent Bopp[1], Olivier Aumont[2], Lester Kwiatkowski[2], Corentin Clerc[1], Léonard Dupont[1], Christian Ethé[3], Roland Séférian[4], Alessandro Tagliabue[5]

[1] LMD/IPSL, Ecole Normale Supérieure/Université PSL, CNRS, Ecole Polytechnique, Sorbonne Université, Paris, France
[2] LOCEAN/IPSL, Sorbonne Université, CNRS, IRD, MNHN, Paris, France
[3] IPSL, Sorbonne Université, CNRS, Paris, France
[4] CNRM, Université de Toulouse, Météo-France, CNRS, Toulouse, France
[5] School of Environmental Sciences, U. Liverpool, Liverpool, UK

*Correspondence to*: Laurent Bopp (bopp@lmd.ipsl.fr)

**Abstract.** The impact of anthropogenic climate change on marine net primary production (NPP) is a reason for concern because changing NPP will have widespread consequences for marine ecosystems and their associated services. Projections by the current generation of Earth System Models have suggested decreases in global NPP in response to future climate change, albeit with very large uncertainties. Here, we make use of two versions of the Institut Pierre Simon Laplace Climate Model (IPSL-CM) that simulate divergent NPP responses to similar high-emission scenarios in the 21st century and identify nitrogen fixation as the main driver of these divergent NPP responses. Differences in the way N-fixation is parameterized in the marine biogeochemical component PISCES of the IPSL-CMs lead to N-fixation rates that are either stable or double over the course of the 21st century, resulting in decreasing or increasing global NPP, respectively. An evaluation of these 2 model versions does not help constrain future NPP projection uncertainties. However, the use of a more comprehensive version of PISCES, with variable nitrogen-to-phosphorus ratios as well as a revised parameterization of the temperature sensitivity of N-fixation, suggests only moderate changes of global-averaged N-fixation in the 21st century. This leads to decreasing global NPP, in line with the model-mean changes of a recent multi-model intercomparison. Lastly, despite contrasting trends in NPP, all our model versions simulate similar and significant reductions in planktonic biomass. This suggests that projected plankton biomass may be a much more robust indicator than NPP of the potential impact of anthropogenic climate change on marine ecosystems across model.



## 1 Introduction

Net Primary Production (NPP) by marine phytoplankton is responsible for nearly 50% of global carbon fixation (Field et al., 1998), and is the basis of almost all marine food chains, controlling the availability of energy and food for upper trophic

levels. As such, marine NPP sustains most oceanic fisheries (Pauly and Christensen, 1995; Stock et al., 2017) and is considered to be one of the most important ecosystem services that the ocean hosts (Pörtner et al., 2014; Bindoff et al., 2019).

Impacts of anthropogenic climate change on marine NPP are particularly alarming as changing NPP could have widespread consequences for marine food ecosystems and the services they provide. For instance, NPP drives the vitality of marine

ecosystems, biogeochemical cycling and the biological carbon pump. Several modelling studies have used Earth System Models (ESMs) to project the evolution of marine NPP over the 21st century under different global warming scenarios (Bopp et al., 2001; Steinacher et al., 2010; Bopp et al., 2013; Cabré et al., 2015; Laufkötter et al., 2015; Kwiatkowski et al. 2020; Tagliabue et al., 2021). Many of these studies have suggested decreases in global NPP in response to future climate change. For the high-emission scenario RCP8.5, estimates of changes in global NPP based on 10 ESMs used in the Coupled

Model Intercomparison Project 5 (CMIP5) range from -2 to -16% in 2090-2099 as compared to 1990-1999 (Bopp et al. 2013). In the recent Coupled Model Intercomparison Project 6 (CMIP6), ESMs also project in average a decrease in global mean NPP under the high-emission scenario SSP5-8.5, albeit with much larger uncertainties than in CMIP5 (Kwiatkowski et al., 2020; Tagliabue et al., 2021).

Multi model climate change projections have been widely used to assess the potential impact of future climate change on

marine biomass across trophic levels (Lotze et al., 2019), fisheries catch potential (Cheung et al., 2010) and global revenues (Lam et al. 2016), and planktonic diversity (Ibarbalz et al., 2019; Benedetti et al., 2021). Using 6 global ecosystem models and climate projections from 2 CMIP5 Earth System Models, Lotze et al. (2019) have shown for instance that the mean global animal biomass in the ocean, largely driven by the decreasing trend in marine NPP, would decrease by 17 % under high emissions by 2100, corresponding to an average 5% decrease for every 1°C of warming.

Despite being used extensively, including in international assessment reports such as the Special Report on the Ocean and Cryosphere in a Changing Climate (IPCC SROCC, Pörtner et al., 2019) and the Global assessment report on biodiversity and ecosystem services (IPBES, Diaz et al., 2019), these projections of future marine NPP are subject to large uncertainties, as demonstrated by inter-model differences (Frölicher et al., 2016). This is especially the case at the regional level, as shown in the Arctic Ocean (Vancoppenolle et al., 2015), in the Southern Ocean (Leung et al., 2015), and in the tropical oceans

(Kwiatkowski et al., 2017; Tagliabue et al., 2020; Tagliabue et al., 2021). It is also the case for the global trend, with some models of the CMIP6 ensemble (IPSL-CM6A-LR, CNRM-ESM2-1, CESM2s) and others not included in the CMIP ensembles (UVIC model in Taucher and Oschlies, 2011; PlankTOM5.3 model in Laufkötter et al., 2015), simulating increasing global NPP in response to anthropogenic climate change. Even within one specific model, poorly constrained



assumptions around key biological components can drive substantial uncertainty in the projected changes in NPP across the
tropical Pacific (Tagliabue et al., 2020).

The differences between models in projecting future NPP result from numerous factors and are underpinned by the delicate
balance between the processes causing NPP decreases (e.g. stratification-driven declines in nutrient supply and temperature-
driven increases in zooplankton grazing) and NPP increases (e.g. stratification-driven declines in light limitation, transport of
excess nutrients and temperature-driven increases in phytoplankton growth rates) (Doney, 2006; Laufkötter et al., 2015). The
effects of other key processes, such as the potential contribution of atmospheric nitrogen fixation (Riche and Christian, 2018;
Wrightson and Tagliabue, 2020), changing nutrient limitation regimes (Tagliabue et al., 2020) or the impact of ocean
acidification on phytoplankton growth (Dutkiewicz et al., 2015) are even more uncertain and are typically only implicitly
parameterized or ignored in current generation ESMs.

Here, we make use of the newly developed version 6 of the Institut Pierre Simon Laplace Climate Model (IPSL-CM6A-LR,
Boucher et al. 2020), used in the framework of the Coupled Model Intercomparison Project Phase 6 (CMIP6, Eyring et al.,
2016) and compare its projected NPP response to the previous version of the same climate model (IPSL-CM5A-LR,
Dufresne et al., 2013) used in CMIP5. These two model versions differ in many ways (spatial resolution in the ocean and
atmosphere, improved versions of multiple model components), but both use the Pelagic Interaction Scheme for Carbon and
Ecosystem Studies (PISCES) model as their marine biogeochemical component. Note that whereas IPSL-CM5A-LR uses
PISCES-v1 (Aumont and Bopp, 2006), the more recent version 2 (PISCES-v2, Aumont et al., 2015) is used in IPSL-CM6A-
LR.

Whereas both models produce near-identical trend in global SST for comparable high-emission scenarios (RCP8.5 and
SSP5-8.5) over the 21st century (Figure 1a, Table 1), a first comparison of NPP projections shows a striking difference, with
NPP decreasing by 9.1% in IPSL-CM5A-LR in 2080-2099 relative to 1986-2005, whereas it increases by 6.8% in IPSL-
CM6A-LR (Figure 1b, Table 1). The aim of this study is to explore and explain this striking global scale divergence. As
shown on Figure 1, we identify the response of biological N-fixation to anthropogenic climate change as one of the main
differences between these 2 ESMs versions, with N-fixation slightly decreasing in IPSL-CM5A-LR and increasing
significantly in IPSL-CM6A-LR over the 21st century (-9% and +75% from 1986-2005 to 2080-2099, respectively, Figure 1,
Table 1).

**Biogeosciences** Open Access
Discussions
EGU

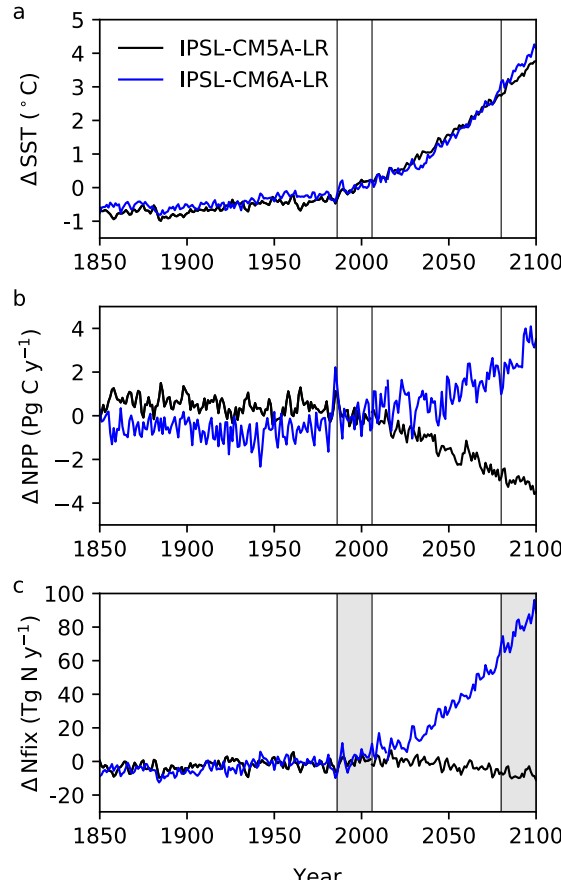

**Figure 1: Simulated changes relative to 1986-2005 in (a) sea surface temperature (°C), (b) integrated net primary production (PgC y-1) and (c) integrated N2 fixation (TgN y-1), for IPSL-CM5A-LR (black) and IPSL-CM6A-LR (blue) over historical and future scenarios. Note that RCP8.5 is used for IPSL-CM5A-LR, but SSP5-8.5 for IPSL-CM6A-LR. The historical and future periods (1986-2005 and 2080-2099, respectively) are displayed as grey boxes.**



Biological dinitrogen ($N_2$) fixation is a key process providing bio-available nitrogen to support marine primary production (Gruber and Galloway, 2008, Zehr and Capone, 2020). Nitrogen fixation was long thought to be confined to the warm, low-latitude ocean, and only performed by very specific cyanobacteria. However, our knowledge has greatly evolved in recent years with the discovery of an ever-increasing array of microorganisms capable of fixing atmospheric nitrogen (Gradoville et



al., 2017). Nitrogen fixation has also been observed in areas where it was previously not thought possible, e.g. in nutrient-rich, low-temperature waters (Tang et al., 2019; Benavides et al., 2018). At present, the response of N-fixation to climate change appears highly unconstrainted, as illustrated by the diversity of responses from ESMs that include some form of N-fixation parameterization (Riche et al., 2018, Wrightson and Tagliabue, 2020). However, despite this variation, because N-fixation emerges rapidly from the background of natural variability, it can be a key driver of NPP trends in N-limited waters

(Wrightson and Tagliabue, 2020).

In this study, we first identify N-fixation as the main process responsible for the sharp contrast between projected NPP in IPSL-CM5A-LR and IPSL-CM6A-LR. We then exploit a series of offline simulations using different versions of the PISCES model (including PISCES-v1, PISCES-v2, and others differing in their representation of N-fixation), consistently forced with the same climate model output. This ensures that no differences in the projections arise from variable climate

scenarios, climate models or model resolution. We then analyse the mechanisms responsible for the different responses of N-fixation in all models used here and discuss these differences in terms of their skill against data-based products. Lastly, we explore the implications of the N-fixation and NPP divergent responses for ocean carbon uptake and for potential impacts on marine ecosystems in the 21st century.

## 2 Material and Methods

### 2.1 Earth System Models

In this study, we make use of two versions of the Institut Pierre Simon Laplace Climate Model (IPSL-CM). The first model, IPSL-CM5A-LR (Dufresne et al., 2013), has been used extensively for the Phase 5 of the Coupled Model Intercomparison Project (CMIP5, Taylor et al., 2011) and compared to other CMIP5 models in terms of its marine biogeochemistry response to climate change in Bopp et al. (2013). The second model is the newly developed IPSL-CM6A-LR (Boucher et al., 2020),

used in the more recent CMIP6 (Eyring et al., 2016) and compared to other CMIP6 models in Kwiatkowski et al. (2020) for the response of marine ecosystem stressors to anthropogenic climate change.

Both IPSL models rely on the same atmospheric (LMDZ), ocean (NEMO) and land surface (ORCHIDEE) model components. However these model components have been substantially revised and upgraded in IPSL-CM6A-LR with respect to IPSL-CM5A-LR. The spatial resolution of the atmospheric model has also been increased from 96x95 points

(mean resolution at 236 km) in longitude and latitude with 39 vertical layers in IPSL-CM5A-LR (Dufresne et al., 2013) to 144x143 points (mean resolution at 157 km) and 79 vertical layers in IPSL-CM6A-LR (Boucher et al., 2020). In addition, the nominal resolution of the ocean model has increased from 2° and 31 vertical layers to 1° and 75 vertical layers.

A detailed description of the changes to LMDZ, NEMO and ORCHIDEE is provided in Boucher et al. (2020). However, we note that the atmospheric general circulation model LMDZ6-A (Hourdin et al., 2020) differs from LMDZ5A (Hourdin et al.

2013) in its inclusion of a new package of parameterizations for turbulence, convection and clouds. The NEMO ocean model comprises 3 components, i.e. ocean dynamics (NEMO-OPA), sea-ice dynamics and thermodynamics (NEMO-LIM) and





marine biogeochemistry (NEMO-PISCES). All of these ocean components have been updated from IPSL-CM5A-LR to IPSL-CM6A-LR, from version 3.2 to version 3.6 of NEMO (Madec et al., 2017; Rousset et al., 2015; Aumont et al., 2015). A detailed assessment of the key properties of the ocean and marine biogeochemical models as used in ESMs that have

contributed to CMIP5 and CMIP6 (including IPSL models) is available in Séférian et al. (2020)

These two versions from the IPSL climate model family qualify as Earth System Models as they include carbon cycle components for the land biosphere (ORCHIDEE) and the ocean (NEMO-PISCES). In the following we describe the PISCES model versions used in this study.

### 2.2 Marine biogeochemical components and N-fixation parameterizations

NEMO-PISCES is an ocean biogeochemical model that simulates marine biological productivity and describes the biogeochemical cycles of carbon, oxygen and of the main limiting nutrients (P, N, Si, Fe). It is based on 24 prognostic tracers in its standard configuration, with 2 phytoplankton functional types (diatoms and nanophytoplankton) and 2 zooplankton size classes (micro- and mesozooplankton). PISCES is by nature a "Redfieldian" model, i.e. it assumes constant stoichiometric ratios for carbon, nitrogen and phosphorus in all organic compartments, but does consider flexible

stoichiometry for iron and silica.

PISCES-v1 is used in IPSL-CM5A-LR and described in details in Aumont and Bopp (2006), whereas PISCES-v2 is used in IPSL-CM6A-LR and described in Aumont et al. (2015). Despite being similar in terms of their overall architecture and number of prognostic tracers, PISCES-v2 differs from PISCES-v1 with an improved representation of iron cycling, phytoplankton growth and nutrient limitation, zooplankton grazing, the sinking of particles, external sources of nutrients, and

the treatment of water-sediment interactions.

In both versions, N-fixation is represented implicitly as a source of ammonium, i.e. without an explicit diazotroph plankton functional type (Figure 2). N-fixation is restricted to warm-waters (T > 20°C) and increases exponentially with temperature following a Q-10 value of 1.9 as for all autotrophic processes in PISCES (Aumont and Bopp, 2006; Aumont et al., 2015). N-fixation is limited by the availability of light and iron, and favoured in low-nitrogen ($NO_3$ and $NH_4$) environments. PISCES-

v1 and PISCES-v2 differ in their treatment of phosphorus limitation on N-fixation, which is not included in PISCES-v1 but combined with iron limitation in PISCES-v2. Finally, due to the fixed stoichiometric ratios between carbon, nitrogen and phosphorus in all organic components it is assumed that N-fixation is accompanied by a release of inorganic phosphorus to account for the fact that diazotrophy-derived organic matter is much richer in N than the standard Redfield assumptions in the model. For every mole of $N_2$ fixed by diazotrophy and instantaneously transferred into the ammonium pool, an additional

0.04 moles of phosphorus is added in the phosphate pool to represent the subsequent remineralization of the diazotrophy-derived organic matter with an N:P ratio of 46:1 (Figure 2a and b). This additional P-source can be interpreted as deriving from the use of an unresolved dissolved organic phosphorus pool by diazotrophs. Note that the overall P-inventory of the ocean is ensured by an annual restoring of the global mean $PO_4$ concentration to its historical global value computed from the World Ocean Atlas 2001 for PISCES-v1 and World Ocean Atlas 2013 for PISCES-v2.



In this work, we use two advanced versions of PISCES-v2. First, PISCES-quota is a newly developed version of PISCES, which accounts for flexible C:N:P stoichiometry. This is accompanied by the introduction of a new plankton functional type (picophytoplankton) and leads to a subsequent increase in the number of prognostic variables to 39 (compared to 24 in all other PISCES versions). A detailed description of PISCES-quota, is provided in Kwiatkowski et al. (2018). As in PISCES-v1 and PISCES-v2, N-fixation is parameterized implicitly, limited to waters with high light levels, low nitrogen, and

adequate iron and phosphorus. However because PISCES-quota is non-Redfieldian, two major changes have been introduced (Figure 2.d): (1) N-fixation consumes and is limited by the availability of an explicit dissolved organic phosphorus pool as evidenced in observations (Sohm and Capone, 2006; Orchard et al., 2010), and (2) the organic matter that is produced by diazotrophy is enriched in nitrogen with respect to phosphorus (with an N:P ratio of 46:1 vs. 16:1 for the canonical Redfield ratio).  In PISCES-quota, implicit diazotrophy transfers nitrogen and phosphorus to three pools: particulate organic matter,

dissolved organic matter, and ammonium and dissolved inorganic phosphorus, with a ratio of one third each (Figure 2.d). Importantly, the temperature effect has been changed following Breitbarth et al. (2007), with a bell-shape response curve and a maximum N-fixation rate set at a thermal optimum of ~ 26°C (Figure 2.e).

Last, and specifically for this study, we modified a version of PISCES-v2 in which only the parameterization of N-fixation was changed based on PISCES-quota (PISCES-v2fix, Figure 2.c). This newly developed parameterization is inspired by

PISCES-quota, except it assumes that the diazotrophy-produced material follows the Redfield stoichiometric ratio of N:P (i.e., 16:1), avoiding any release of additional inorganic phosphorus as in PISCES-v1 and PISCES-v2 (Figure 2). PISCES-v2fix uses the same temperature-dependency as PISCES-quota (Figure 2.e).

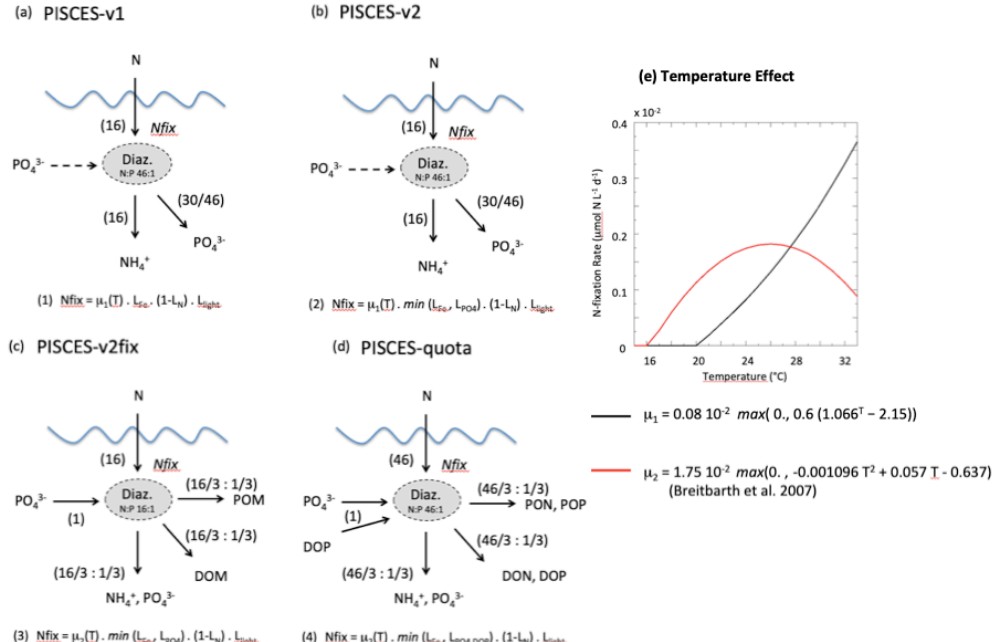

**Figure 2: Schematic diagram describing the parameterization of N-fixation for each of the PISCES versions used in this study. Equations (1) to (4) give N-fixation rates as a function of μ the N-fixation rate at temperature T, $L_{PO4}$, $L_{Fe}$ and $L_N$ limitation terms (varying between 0 and 1) for phosphorus, iron and nitrogen, respectively, and $L_{light}$ a light limitation term (also between 0 and 1). The values in parenthesis denote the number of moles (of nitrogen or phosphorus) that are consumed and produced by the implicit N-fixation parameterization. (e) N-fixation rate (in $\mu molN \, L^{-1} \, d^{-1}$) as a function of water temperature and in the case of no other limitation, for PISCES-v1 and PISCES-v2 (black curve) and PISCES-v2fix and PISCES-quota (red curve, from Breitbarth et al. 2007).**

## 2.3 Simulations

IPSL-CM5A-LR and IPSL-CM6A-LR have been integrated following the CMIP5 (Taylor et al., 2012) and CMIP6 (Eyring et al., 2016) protocols, respectively. After long spin-up integrations in pre-industrial conditions, both Earth System Models are run under historical forcing (from 1850 to 2005 or 2014), and then under high-emission scenarios (from 2006 or 2015 to 2100, following RCP8.5 for IPSL-CM5A-LR and SSP5-8.5 for IPSL-CM6A-LR). Details on historical and projection simulations with IPSL-CM5A-LR and IPSL-CM6A-LR are given in Dufresne et al. (2013) and Boucher et al. (2020), respectively.





In addition to these coupled Earth System simulations and to facilitate the assessment of the role of biogeochemical
parameterizations, we performed a series of "offline" ocean biogeochemistry simulations under the same physical forcing.
The three PISCES versions that were run offline are PISCES-v2, PISCES-v2fix and PISCES-quota. To do so, we used the
physical output of IPSL-CM5A-LR under historical and RCP8.5 conditions (monthly means of ocean temperature, salinity,
currents, and mixed layer-depth) and forced these different PISCES versions over 1850-2100, so that the only differences
between the offline simulations originate from the biogeochemical parameterizations.

In the latter, the coupled simulations are referred to as IPSL-CM5A and IPSL-CM6A, whereas the offline simulations are
referred as PISCES-v2, PISCES-v2fix and PISCES-quota. To facilitate the comparison of simulations run under different
protocols, the same 20-yr periods have been retained for historical (1986-2005) and future (2080-2099) baseline periods.

### 2.4 Cluster Analysis

To spatially relate changes in nitrogen fixation with changes in phytoplankton growth, we used a clustering analysis based
on mutual information as a simple way of assessing the statistical dependence of our variables. For both IPSL-CM5A and
IPSL-CM6A, we computed vertically-integrated anomalies for nitrogen fixation and phytoplankton-realised growth ($\mu$)
between the 2080-2099 and the 1986-2005 periods and normalised them between -1 and 1. Since phytoplankton growth rates
were not directly available model outputs, we used the ratio of biomasses and NPP to recompute them. The two anomalies
were then discretised in twenty, evenly-spaced bins. For each combination of anomalies ($X_{Nfix},Y_{\mu}$) from the discrete set, we
compute the pointwise mutual information (PMI, Cover and Thomas, 1991) as follows:

$$\textbf{PMI}(X_{Nfix},Y_{\mu}) = \log(\ \textbf{P}(X_{Nfix},Y_{\mu})/(\textbf{P}(X_{Nfix})\textbf{P}(Y_{\mu}))\ )$$

The resulting PMI-matrix (Figure 4, top row) was then split into a set of sensible clusters, based on the amount of shared
information between our variables and the sign of the anomalies. Following this pipeline, we could both (i) visually assess
differences between IPSL-CM5A and IPSL-CM6A behaviours looking at the matrix and (ii) further project these clusters on
2D maps, to analyse their spatial distributions. Only anomaly combinations with an absolute PMI greater than 0.01 were
considered.

### 3 Results & Discussions

#### 3.1 Climate change driven responses of NPP and N-fixation in IPSL Earth System Models

We compare here two successive generations of the IPSL Climate Model, IPSL-CM5A (Dufresne et al., 2013) and IPSL-
CM6A (Boucher et al., 2020), forced over the 21st century by two similar high-emission scenarios (RCP8.5 and SSP5-8.5).
The Equilibrium Climate Sensitivity (ECS) has increased slightly, from 4.1 K in IPSL-CM5A to 4.8 K in IPSL-CM6A
(Boucher et al., 2020), both values being in the upper range of ECS from climate models developed in the framework of





CMIP5 and CMIP6 (Forster et al., 2019). As a consequence, the warming level for the global mean sea surface temperature

under the high-emission scenarios is slightly higher in IPSL-CM6A (+3.57 °C) than in IPSL-CM5A (+3.27 °C) at the end of
the 21st century.

| Model Version | SST °C | Nfix TgN y$^{-1}$ | NPP (glob) PgC y$^{-1}$ | NPP (90°S-30°S) PgC y$^{-1}$ | NPP (30°S-30°N) PgC y$^{-1}$ | NPP (30°N-90°N) PgC y$^{-1}$ |
|---|---|---|---|---|---|---|
| IPSL-CM5A | +3.27 | -7.3 (-9.0%) | -3.1 (-9.1%) | -0.1 | -2.6 | -0.4 |
| IPSL-CM6A | +3.57 | +77.9 (+75%) | +2.9 (+6.8%) | +0.6 | +1.9 | +0.4 |
| PISCES-v2 | +3.27 | +76.1 (+68.1%) | +6.6 (+13.8%) | +1.0 | +5.1 | +0.5 |
| PISCES-v2fix | +3.27 | +15.6 (+20.1%) | -0.7 (-1.6%) | +0.9 | -1.9 | +0.3 |
| PISCES-quota | +3.27 | +6.0 (+6.5%) | -2.2 (-5.7%) | +0.3 | -2.6 | +0.1 |

**Table 1: Climate change impact on global mean sea surface temperature (°C), depth integrated N-fixation (TgN y$^{-1}$) , and depth
integrated net primary production (PgC y$^{-1}$). All are absolute differences between 2080-2099 and 1986-2005, with relative changes
in parenthesis.**

As stated in the introduction, the two climate models that we compare here simulate opposing projections of global oceanic
NPP over the 21$^{st}$ century. In IPSL-CM5A, NPP decreases by 3.1 PgC y$^{-1}$ from 1986-2005 to 2080-2099 (under RCP8.5),
whereas IPSL-CM6A simulates an increase of NPP by 2.9 PgC y$^{-1}$ over the same period (but under SSP5-8.5) (Table 1,
Figure 1). These NPP global changes represent a 9.1% decrease and a 6.8% increase for IPSL-CM5A and IPSL-CM6A,
respectively.

The difference between the two model versions mostly arises from the tropical oceans (30°S-30°N), with a decrease of 2.6
Pg y$^{-1}$ (-14.3%) and an increase of 1.9 Pg y$^{-1}$ (+7.3%) in IPSL-CM5A and IPSL-CM6A, respectively (Table 1). The extra-
tropical oceans also show notable differences between the two model versions, but these differences contribute much less to
the global NPP changes (Table 1). At the regional scale, the largest differences between the two model versions are located
in the oligotrophic gyres of all ocean basins, which show significant increases in NPP for IPSL-CM6A (up to +45 gC m$^{-2}$ y$^{-1}$)
but slight decreases or increases in IPSL-CM5A (Figure 3). Elsewhere, the general patterns are similar across the model
versions with NPP increasing at high latitudes and decreasing in the equatorial band and at the poleward borders of
subtropical gyres (Figure 3).

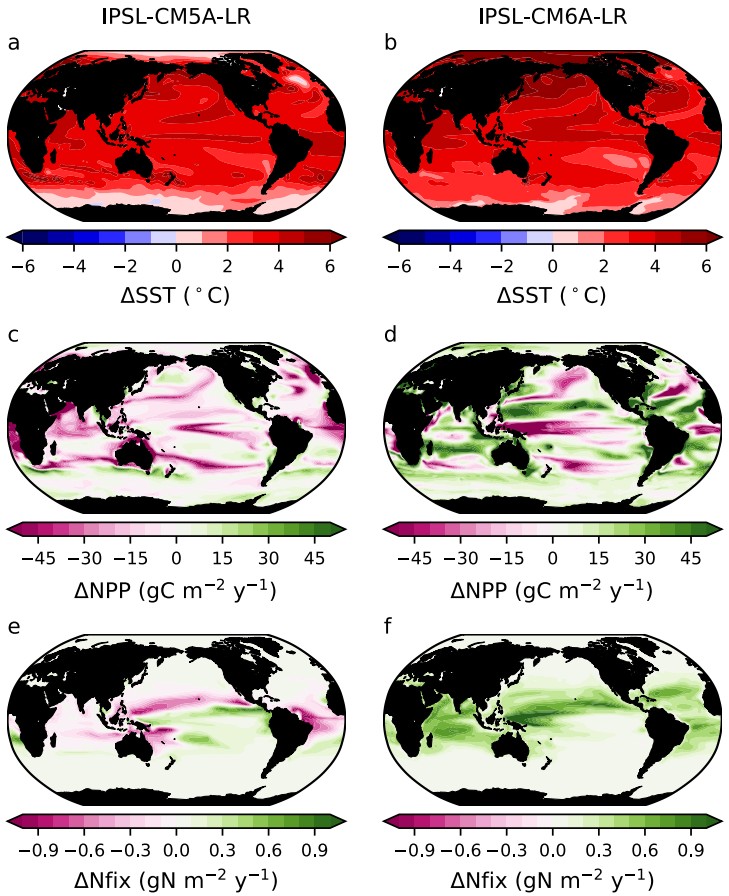

**Figure 3: Changes in (a,b) sea surface temperature (°C), (c,d) vertically-integrated net primary production (gC m⁻² y⁻¹) and (e,f)**
**vertically-integrated N-fixation (gN m⁻² y⁻¹) between 1986-2005 and 2081-2100. Left panels are for IPSL-CM5A under RCP8.5,**
**whereas right panels are for IPSL-CM6A under SSP5-8.5**

The localization of NPP differences in the oligotrophic gyres, and the similarity of the ocean physics response (not shown) to

anthropogenic climate change, point towards a role for nitrogen fixation in explaining the contrast between IPSL-CM5A and

IPSL-CM6A in terms of NPP projections. Indeed, whereas N-fixation slightly decreases in IPSL-CM5A (-7.3 TgN y⁻¹; -9%)

through the 21st century, it almost doubles in IPSL-CM6A (+77.9 TgN y⁻¹; +75%) (Figure 1, Table 1).





Spatially, the regions in which the increases in N-fixation are strongest coincide with the regions in which NPP also increases strongly in IPSL-CM6A (Figure 3). To gain a better insight into how the two variables co-vary in IPSL-CM5A and IPSL-CM6A, we computed the point-mutual information (Cover and Thomas, 1991) between phytoplankton growth-rate

anomalies and N-fixation anomalies, as a probabilistic measure of correlation (see Methods, Figure 4). The 5 clusters that emerge are referred to according to the co-variations of our 2 variables, i.e. phytoplankton growth rate anomalies and N-fixation anomalies, with cluster 1 – red (cluster 3 – yellow) when both variables decrease (increase) with time, cluster 2 – blue (cluster 5 – green) when phytoplankton growth rates increase and N-fixation decreases (when phytoplankton growth rates decrease and N-fixation increases), and cluster 4 – pink when growth rates increase without any significant change in

N-fixation. While the surface occupied by the pink cluster (increases in growth rate, no significant change in N-fixation) remains the same in the 2 models, nearly all IPSL-CM5A green and red cluster points shift to yellow in IPSL-CM6A (Figure 4). In other words, regions where growth rate changes were independent from N-fixation in IPSL-CM5A, correspond to regions where phytoplankton growth rates and diazotrophy are positively correlated in IPSL-CM6A. As expected, areas where NPP decreases are independent from N-fixation (blue cluster, e.g. the North Atlantic) were similar in both runs. This

analysis confirms the reinforced relationship between phytoplankton growth and diazotrophy as a key difference between the IPSL-CM6A and IPSL-CM5A models.



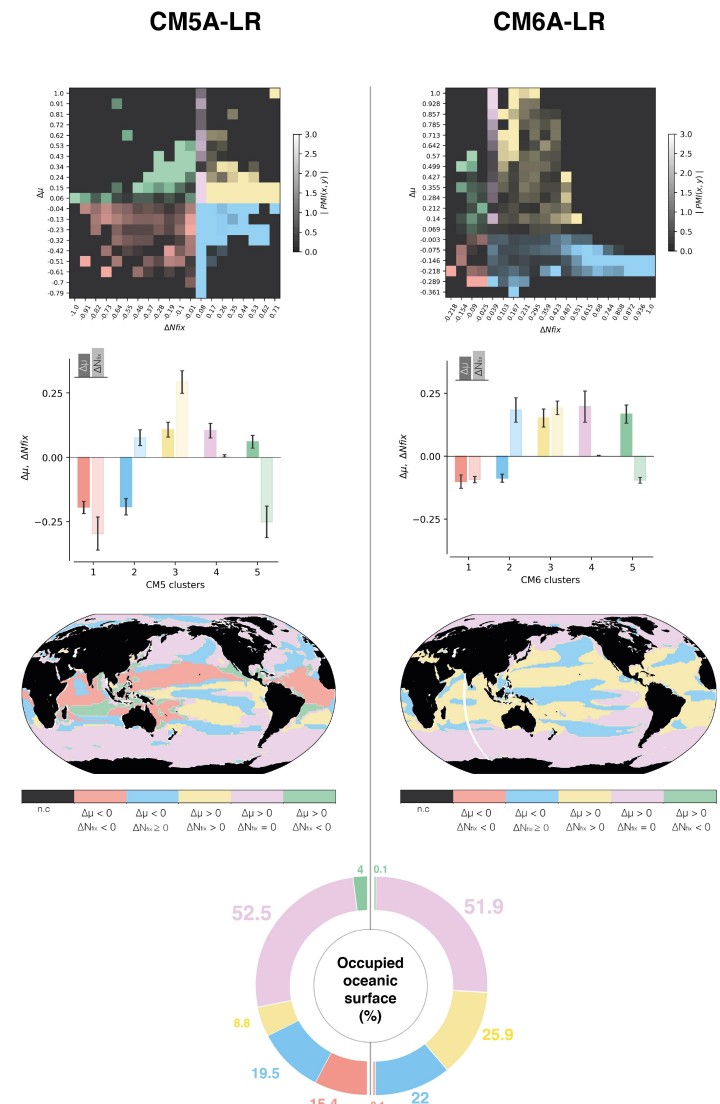




**Figure 4: Cluster analysis of phytoplankton growth rates (m) and N-fixation (Nfix) changes in IPSL-CM5A and IPSL-CM6A. All changes refer to 2080-2099 as compared to 1986-2005. (a) Pointwise-mutual information between N-fixation (x-axis) and growth rates (y-axis). The matrix was split into five clusters based on combinations of anomaly values (quadrants). (b) Bar-plots corresponding to mean values of the anomalies, for each of the defined clusters. (c) Robinson projections of the clustered points.**

**(d) Half-pie charts, representing the proportion of oceanic surface occupied by each cluster.**

**3.2 Climate change driven responses of NPP and N-fixation in PISCES offline simulations**

To further understand the role of N-fixation in explaining the differences between IPSL-CM5A and IPSL-CM6A, we use additional offline simulations where the same ocean physical forcing is applied to different versions of the PISCES biogeochemical model. This enables a direct comparison of specific biogeochemical parameterizations within the same

physical framework (see Methods 2.3).

By using the same PISCES version (PISCES-v2; Aumont et al., 2015) as that in IPSL-CM6A, but forced with the physical output of IPSL-CM5A under RCP8.5 (simulation PISCES-v2, see Section 2.3), we obtain a very similar response to that in IPSL-CM6A in terms of N-fixation, with an increase of 76.1 TgN y$^{-1}$ (+68.1%) from 1986-2005 to 2080-2099, as compared to 77.9 TgN y$^{-1}$ (+75%) over the same period in IPSL-CM6A (Table 1, Figure 5). This shows that the differences between

IPSL-CM6A and IPSL-CM5A are robust in a common physical framework. Consequently, NPP also increases in PISCES-v2 (by +13.8%) (Table 1, Figure 5). As in IPSL-CM6A, the increase in NPP in this offline simulation is largely concentrated in the tropical oceans, where it reaches + 17% (Table 2), and is typically coincident with regions of increasing N-fixation (Figure 5).

In the PISCES-v2fix simulation, the temperature dependency of N-fixation and the assumed N:P ratio of the implicit

diazotrophs is modified from that in PISCES-v2 (Section 2.1), while the same forcing, that derived from IPSL-CM5A, is applied (Section 2.3). In this offline simulation, N-fixation also increases over the 21st century (by 15.6 TgN y-1 or 20.1% at the end of the 21st century), but much less than in PISCES-v2. The increase in N-fixation is dampened everywhere as compared to PISCES-v2, with even regions where N-fixation decreases in PISCES-v2fix (west tropical Pacific, tropical Atlantic) (Figure 5). Consequently, NPP changes are markedly different than those from PISCES-v2, with a global decrease

of 1.6% in 2080-2099 (Table 1), mostly located in the tropics (Figure 5).

Lastly, we also compare these offline simulations with an offline simulation of the newly developed PISCES-quota model (Kwiatkowski et al., 2018), an advanced version of PISCES-v2 in which the assumption of fixed phytoplankton C:N:P stoichiometry has been relaxed. In this simulation, N-fixation only slightly increases (by 6 TgN y-1 or 6.5% over the 21st century) whereas NPP decreases (by 2.2 PgC y-1 or 5.7%) over the 21st century. At regional scales, the patterns of N-

fixation display contrasting tendencies with decreases in the western Pacific, equatorial Atlantic and Indian Ocean and increases poleward of these regions. The regional changes of NPP resemble those simulated in IPSL-CM5A (Figure 5). The





differences between PISCES-quota and the two other offline simulations (PISCES-v2 and PISCES-v2fix) originate from the major developments in PISCES-quota such as variable C:N:P stoichiometry and the inclusion of a third phytoplankton functional type (picophytoplankton) as demonstrated in Kwiatkowski et al. (2018).

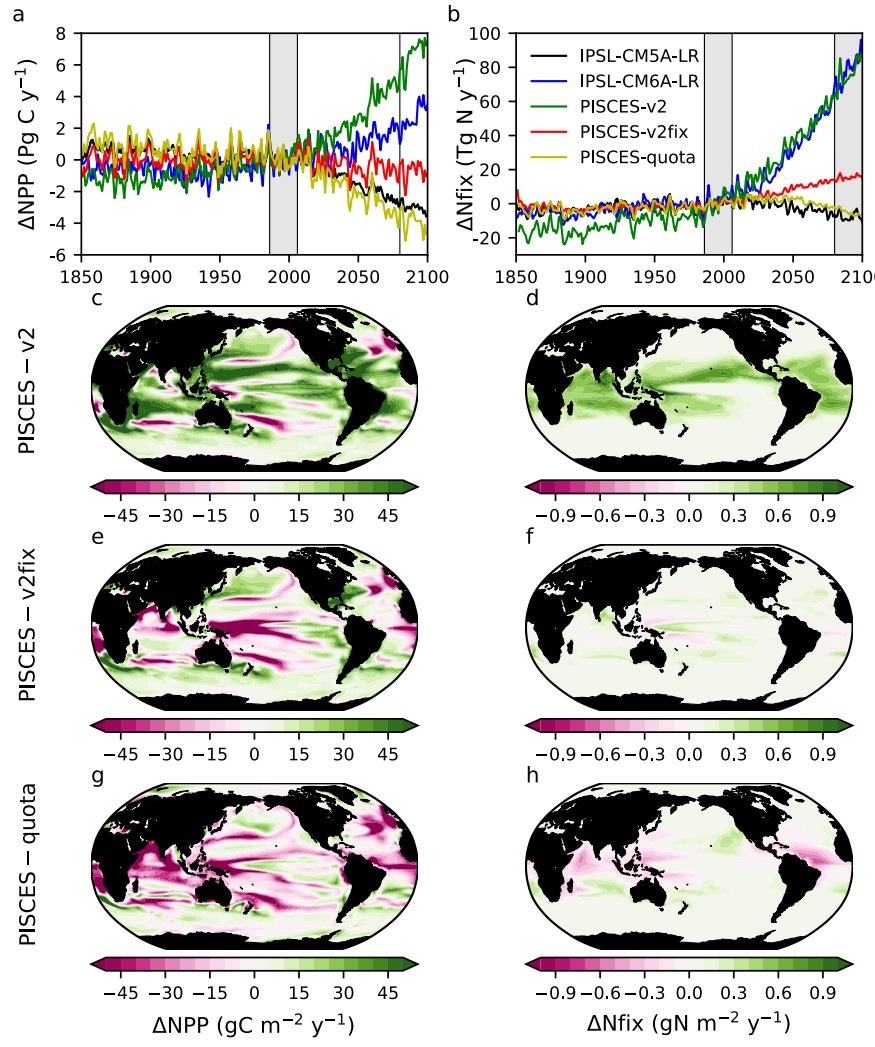


**Figure 5: Changes in NPP and N-fixation across all model versions.** Time series of (a) global N-fixation anomalies (TgN y$^{-1}$) and (b) global NPP anomalies (PgC y$^{-1}$), relative to 1986-2005 and under historical, RCP8.5 (all model versions except IPSL-CM6A-LR)



**and SSP5-8.5 (only IPSL-CM6A-LR). Changes in (c,e,g) vertically-integrated N-fixation (gN m$^{-2}$ y$^{-1}$) and in (d,f,h) vertically-integrated NPP (gC m$^{-2}$ y$^{-1}$) for all offline model versions in 2080-2099 relative to 1986-2005, under RCP8.5.**


The comparison of our 3 offline simulations demonstrates the role of the response of N-fixation in the evolution of NPP over the 21st century. Under the same physical forcing, N-fixation increases by +68.1%, +20.1% and +6.5% over the 21st century, in PISCES-v2, PISCES-v2fix, and PISCES-quota, respectively. The resulting changes in global NPP over the same period are +13.8%, -1.6% and -5.7%, in the same 3 versions, respectively. The comparison of these offline simulations

clearly links the response of global NPP to the parameterization of N-fixation, and hence reinforces the assumption that differences in N-fixation are the primary driver of the divergent IPSL-CM5A and IPSL-CM6A NPP projections.

### 3.3 Mechanisms determining the N-fixation response

The contrasting responses of N-fixation in the IPSL-CM5A and IPSL-CM6A models (and thus between PISCES-v1 and PISCES-v2) is not entirely intuitive, as the 2 models share very similar parameterizations of N-fixation (e.g. the same N-

fixation temperature sensitivity) (Figure 2). To explain these contrasting trends, we focus on an area located in the northwestern tropical Pacific (130°E-160°E, 10°N-20°N), where the response of nitrogen fixation diverges strongly between IPSL-CM5A and IPSL-CM6A (Figure 6a) and exploit the comparison between IPSL-CM5A and the offline versions of PISCES (PISCES-v2, PISCES-v2fix and PISCES-quota) that use an identical climate forcing. In this region N-fixation increases by 33% in PISCES-v2 between 1986-2005 and 2081-2100 (of the same order of magnitude as in IPSL-CM6A,

+45%, not shown), decreases by 85% in IPSL-CM5A, and decreases by 2% and 10% in PISCES-v2fix and PISCES-quota, respectively (Figure 6b). Concurrently, sea surface temperature in this region increases by nearly 4°C (between 1986-2005 and 2081-2100) to reach 31.5°C at the end of the 21st century. In IPSL-CM5A and PISCES-v2, this increase leads to a boost in N-fixation by a factor of 2.1 (Figure 6d). In PISCES-v2fix and PISCES-quota, on the contrary, the increase in temperature reduces N-fixation by more than 30% (Figure 6d) because of the bell-shape T-response function from Breitbarth et al.

335  (2007).

In PISCES-v2, the limitation terms due to light, phosphate, iron and excess nitrate remain inoperative and N-fixation responds almost exclusively to temperature (Figure 6b). In IPSL-CM5A, on the other hand, the continuous increase in nitrate concentration (Figure 6e), and the shift towards positive N* values (Figure 6h) leads to a limitation or even a decrease in N-fixation due to the $L_N$ term (limitation by excess inorganic nitrogen) (Figure 2b, equation (2)). In IPSL-CM5A, NO$_3$

concentrations remain close to zero, N* values remain slightly negative, and the $L_N$ term has no influence, allowing N-fixation to increase in response to temperature. As in IPSL-CM5A, in PISCES-v2fix and PISCES-quota, NO$_3$ concentrations also remain close to zero (Figure 6e), the excess nitrogen limitation term has no influence, and N-fixation decreases slightly due to warming (Figure 6b).

In summary, we show that the N-fixation response is highly sensitive to (1) the parameterization used for the temperature

sensitivity (PISCES-v2 as compared to PISCES-v2fix and PISCES-quota), and (2) the respective evolution of NO$_3$ and PO$_4$

concentrations (IPSL-CM5A as compared to PISCES-v2). To reiterate, it is the divergent responses of N-fixation in oligotrophic gyres (Figure 6a), where temperatures can exceed the optimal values in Breitbarth et al. (2007)'s parameterization (Figure 2), and where $NO_3$ accumulation can induce a decrease in the rate of N-fixation, that explain the differences between the different versions of PISCES.

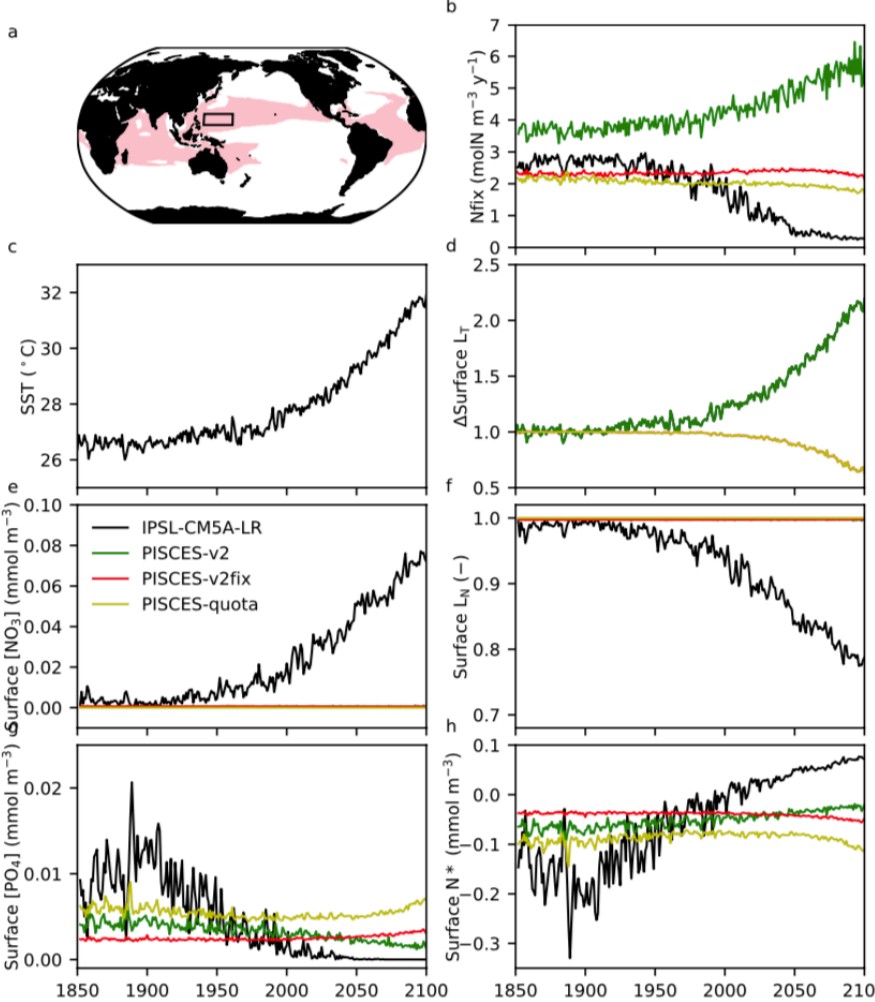


**Figure 6: Mechanisms explaining the contrasting response of N-fixation across the different PISCES versions. (a) Regions where IPSL-CM5A and IPSL-CM6A simulate an N-fixation response of opposing sign (red), and where the detailed analysis is performed (130°E-160°E, 10°N-20°N, black box). (b) N-fixation (molN m$^{-3}$ y$^{-1}$) in the different PISCES versions. (c) Sea surface**



temperature (°C) in IPSL-CM5A and (d) relative change in surface $L_T$ (no unit) for IPSL-CM5A and PISCES-v2 (black-green),
PISCES-v2fix and PISCES-quota (red-yellow). (e) Surface $NO_3$ concentration (mmol m$^{-3}$), (f) surface $L_N$ (no unit), (g) surface $PO_4$
concentration (mmol m$^{-3}$) and (f) surface N* (mmol m$^{-3}$) in all PISCES versions. All (from b to h) are annual-mean time-series
averaged over 130°E-160°E and 10°N-20°N.

### 3.4 Evaluation and constrains on projections

As the responses of the different versions of PISCES to anthropogenic climate change were so variable (see Section 3.1 and
3.2), we conducted a brief evaluation of these simulations over the historical period for NPP, nutrients (nitrate, phosphate),
and N-fixation to determine if any of the PISCES versions have significantly better performance scores than the others (see
methods). We note that the outputs of IPSL-CM5A and IPSL-CM6A are evaluated in Séférian et al. (2020) alongside the
other Earth System Models of the CMIP5 and CMIP6 exercises.

A visual inspection of the bias maps for NPP, excess N (defined here as N* = $NO_3$ – 16*$PO_4$), and N-fixation fails to
highlight a single model version that outperforms the others, with similar regional biases for all PISCES variants (Figure 7).
All versions tend to underestimate NPP in the mid-latitudes of the Northern Hemisphere, and overestimate NPP in the
equatorial band and the Southern Ocean. For N*, the regional biases are also similar, with an underestimation in the
Southern Ocean and a marked overestimation in the North Pacific. Finally, comparison of simulated N-fixation rates to the

measured estimates of Landolfi et al. (2018) show a fairly widespread underestimation, for all versions of PISCES, in the
northern subtropical gyre of the Pacific and in the equatorial Atlantic.

A more quantitative analysis using RMSE for each of the above fields confirms this visual impression with very similar
RMSEs across all versions of PISCES. It can be noted, however, that the distribution of N* and NPP seem to be better
reproduced in PISCES-quota, while the comparison of N-fixation rates gives the worst scores for PISCES-quota.

In conclusion this brief comparison with observations fails to distinguish between the different versions of the PISCES
model used here. To go further, it would be necessary to compare the simulated trends over the historical period with time
series obtained at marine stations (e.g. Hawaii Ocean Time-series programme, Karl et al., 2014; Bermuda Atlantic Time-
Series, Lomas et al., 2013), or with reconstructions of the evolution of N-fixation over the last century from
paleoceanographic proxies such as $\delta^{15}$N (Sherwood et al., 2014).




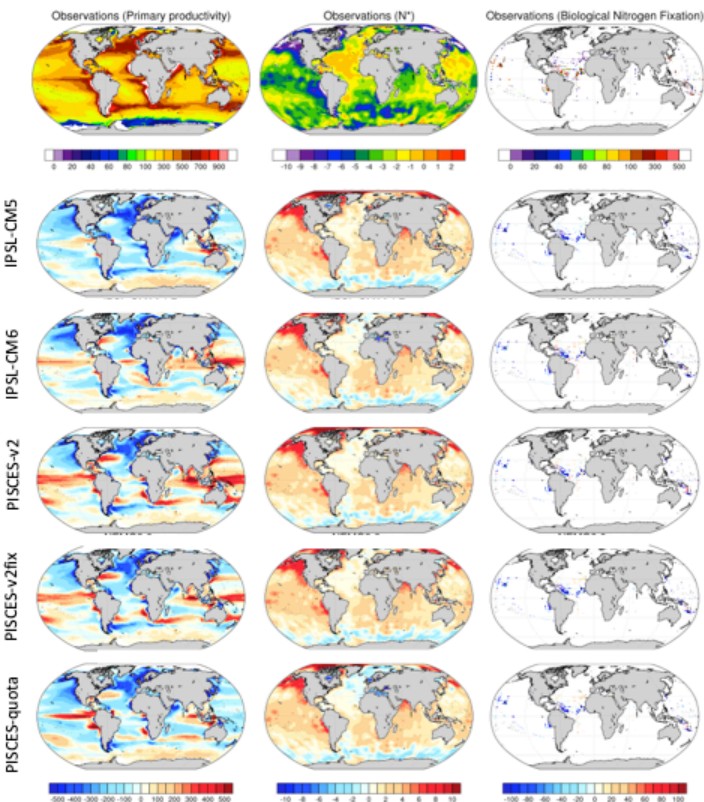

**Figure 7: Model-data intercomparison of NPP (mgC m$^{-2}$ d$^{-1}$), N* (mmol m$^{-3}$) and N-fixation (mmolN m$^{-2}$ y$^{-1}$). The upper panels show NPP based on remote-sensing observations with the VGPM algorithm (Behrenfeld et al., 2005), N* from NO$_3$ and PO$_4$ as provided in the World Ocean Atlas Database (Garcia et al. 2014), and N-fixation from Luo et al. (2014) updated by Landolfi et al.**
**(2018). The other panels show model-data anomalies averaged over the period 1986-2005.**

| Model Version | NPP (mgC m$^{-2}$ d$^{-1}$) | N Fixation (mmolN m$^{-2}$ y$^{-1}$) | N* (mmol m$^{-3}$) |
|---|---|---|---|
| IPSL-CM5 | 1.936 | 24.103 | 2.028 |
| IPSL-CM6 | 1.730 | 26.004 | 1.868 |





| | | | |
|---|---|---|---|
| PISCES-v2 | 1.519 | 22.671 | 2.134 |
| PISCES-v2fix | 1.621 | 41.824 | 2.172 |
| PISCES-quota | 1.478 | 48.120 | 1.300 |

**Table 2: Root-mean square errors (RMSE) for NPP (mgC m$^{-2}$ d$^{-1}$), N-fixation (mmolN m$^{-2}$ y$^{-1}$) and N\* (mmol m$^{-3}$) for all PISCES**
**versions used here against observations (NPP based on remote-sensing (Behrenfeld et al., 2005), N\* from NO$_3$ and PO$_4$ as provided in the World Ocean Atlas Database (Garcia et al., 2014), and N-fixation from Luo et al. (2014) updated by Landolfi et al. (2018). All model-estimates are for 1986-2005.**

Emergent constraints relate observable trends or sensitivities across a model ensemble to future differences in model
simulations in order to constrain projection uncertainties (Allen and Ingram, 2002; Hall and Qu, 2006; Hall et al., 2019).
They have been used extensively within the earth sciences to constrain projections as diverse as climate sensitivity (Caldwell
et al., 2018), snow albedo feedbacks (Hall and Qu, 2006), precipitation extremes (O'Gorman, 2012; DeAngelis et al., 2016)
and carbon cycle feedbacks (Cox et al., 2013; Wenzel et al., 2014; Goris et al., 2018; Terhaar et al., 2020).

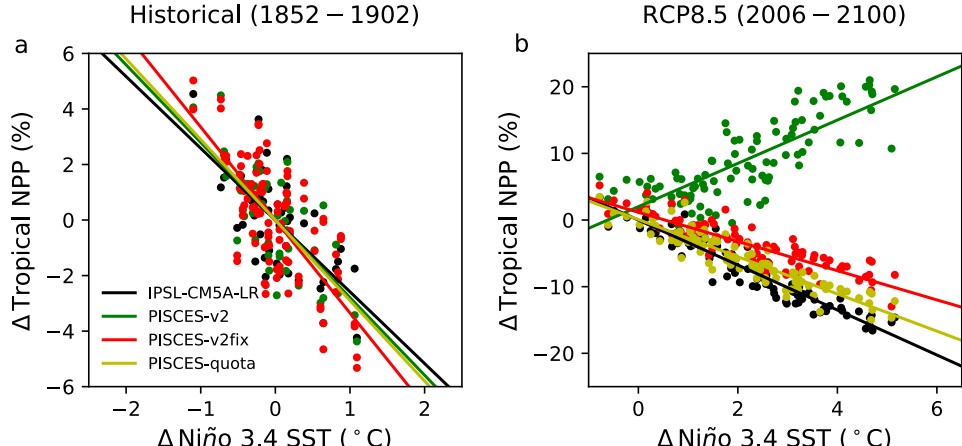


**Figure 8: The relationship between annual anomalies in tropical (30°N-30°S) NPP and Niño3.4 region SST anomalies over a, 1852-1902 of historical simulations and b, 2006-2100 of RCP8.5 for IPSL-CM5A (black), PISCES-v2 (green), PISCES-v2fix (red) and PISCES-quota (yellow).**





Kwiatkowski et al., (2017) applied an emergent constraint approach to CMIP5 NPP projections, finding an emergent relationship between the tropical (30°N-30°S) sensitivity of integrated NPP to El Niño Southern Oscillation (ENSO) variability (Niño 3.4 SST anomalies) in pre-industrial control simulations and the tropical NPP response to 21$^{st}$ century warming under a high-emissions scenario (RCP8.5). That is, models that exhibited heightened tropical NPP sensitivity to ENSO-driven SST fluctuations, typically exhibited greater future NPP declines in response to climate change. The authors
then used observational estimates of the NPP-ENSO sensitivity to constrain the future NPP response.

The present study highlights that N-fixation may represent a slow or threshold process that has the potential to limit the applicability of emergent constraint approaches to projections of NPP. Indeed, the NPP-ENSO sensitivities of IPSL-CM5A, PISCES-v2 and PISCES-v2fix are shown to be highly similar in the first 50 years of historical simulations (-2.6 % °C$^{-1}$, -2.8 % °C$^{-1}$ and -3.3 % °C$^{-1}$) and yet the twenty-first century NPP sensitivity of PISCES-v2 is dramatically changed (+3.2 % °C$^{-1}$
$^{-1}$), due to the enhanced role of N-fixation, while the sensitivity of IPSL-CM5A and PISCES-v2fix remain consistent (-3.4 % °C$^{-1}$ and -2.2 % °C$^{-1}$) (Figure 8). The extent to which the differing PISCES-v2 model behaviour across time scales challenges the previously identified emergent constraint is unknown and it is also possible that future changes move beyond historical variability (e.g. Tagliabue et al., 2020). Interestingly, the tropical NPP sensitivity of PISCES-quota in the early historical (-2.9 % °C$^{-1}$) and under RCP8.5 (-2.8 % °C$^{-1}$) is highly similar and in line with IPSL-CM5A and PISCES-v2fix (Figure 8). As
such, there is strong reason to doubt whether N-fixation sensitivity to climate change, and hence the response of NPP, in PISCES-v2, are realistic. However, given all of the above, verifying the validity of the Kwiatkowski et al. (2017) NPP emergent constraint with multiple quota models is identified as a priority. It is urgent as well to verify the sensitivity of nitrogen fixation with more mechanistic models of N fixation (such as that of Pahlow et al. (2013) or Inomura et al. (2018)).

**3.5 Implications for carbon uptake, ocean deoxygenation and impact on plankton biomass**

The last question we explore is the potential implications of this large difference in NPP projections between IPSL-CM5A and IPSL-CM6A on (1) carbon export and anthropogenic carbon uptake, (2) ocean deoxygenation and (3) impacts on plankton biomass.

| Model Version | C uptake (1851-2100, PgC) | Export Corg at 100m (PgC) | Subsurface O$_2$ (mmol m$^{-3}$) | Phyto biomass (TgC) | Zoo biomass (TgC) |
|---|---|---|---|---|---|
| IPSL-CM5A | 472.1 | 7.07 -1.21 (-17.12 %) | 199.4 -8.19 (-4.11 %) | 1014 -67.6 (-6.67 %) | 1039 -155.1 (-14.93 %) |
| IPSL-CM6A | 488.2 | 7.31 -0.166 (-2.27 %) | 190.8 -17.96 (-9.41 %) | 816.1 -35.18 (-4.31 %) | 645.9 -69.06 (-10.69 %) |
| PISCES-v2 | 404.0 | 8.25 0.19 (2.31 %) | 193.6 -15.79 (-8.16 %) | 893.0 -40.04 (-4.48 %) | 765.9 -49.16 (-6.42 %) |
| PISCES-v2fix | 382.6 | 7.70 -0.84 | 195.3 -12.67 | 886.3 -42.42 | 741.4 -86.71 |





| | | (–10.91 %) | (–6.48 %) | (–4.79 %) | (–11.70 %) |
|---|---|---|---|---|---|
| PISCES-quota | 419.6 | 7.05 -0.94 (–13.39 %) | 195.3 -10.71 (–5.49 %) | 826.9 -45.47 (–5.50 %) | 822.0 -94.52 (–11.50 %) |

**Table 3: Ocean Carbon Sink, Impact of climate change on organic matter export at 100m, sub-surface (100-600m) oxygen concentrations, phytoplankton and zooplankton biomasses in the different model versions. All columns indicate absolute values for 1986-2005 and differences between 2080-2099 and 1986-2005 (absolute and relative in parentheses), except for anthropogenic carbon where it is the cumulative anthropogenic carbon uptake over 1851-2100.**

As expected from NPP differences, the export of particulate organic matter at 100m is strongly reduced in IPSL-CM5A (-17.1%) at the end of the 21st century, whereas it is only slightly affected in IPSL-CM6A (-2.3% in 2080-2099 as compared to 1986-2005) (Table 3). The potential impact of this contrasting response of the biological pump in IPSL-CM5A vs. IPSL-CM6A is moderate, with a cumulative ocean carbon sink in 2100 of 472.1 and 488.2 PgC, for IPSL-CM5A and IPSL-CM6A, respectively. Note that the difference between these numbers also reflects the fact that the scenarios (RCP8.5 and

SSP5-8.5) and the oceanic physical response (e.g. warming, stratification and changes in circulation) are different across the 2 ESM versions. When comparing the offline PISCES simulations, in which the scenarios and the physical ocean changes are identical, the cumulative ocean carbon uptake in 2100 is 404, 382.6 and 419.6 PgC in 2100 for PISCES-v2, PISCES-v2fix and PISCES-quota, respectively (Table 3). The difference between PISCES-v2 and PISCES-v2fix, which amounts to 21.4 PgC, is solely due to the change in the N-fixation parameterization utilised, whereas the difference with PISCES-quota

also reflects other biogeochemical changes (e.g. increased C:N and C:P ratios of the produced organic matter as discussed in Kwiatkowski et al., 2018).

For ocean deoxygenation, the intensity of the subsurface signal we obtain with IPSL-CM5A and IPSL-CM6A may indeed be driven by the opposing NPP changes. Whereas subsurface $O_2$ concentrations only decrease by 8.2 mmol m$^{-3}$ (-4.1%) on average in IPSL-CM5A, they decrease by 17.96 mmol m$^{-3}$ (-9.4%) in IPSL-CM6A (Table 3). Spatially, the general patterns

of subsurface ocean deoxygenation are similar across the two models, with the largest decreases, up to -60 mmol m$^{-3}$ in 2080-2099, occurring in the North Atlantic, North Pacific and in the Southern Ocean (not shown). In the tropical oceans, the changes are weaker, from -20 to +20 mmol m$^{-3}$ in IPSL-CM5A and from -20 to +10 mmol m$^{-3}$ in IPSL-CM6A. Note that the regions where subsurface $O_2$ concentrations increase in IPSL-CM5A, such as in the tropical Indian and Atlantic oceans, as discussed in Bopp et al. (2017), display on the contrary decreasing trends in IPSL-CM6A. We interpret this difference as due

to the response to increasing NPP, increasing export of organic matter and the subsequent remineralization-driven consumption of $O_2$ at depth in IPSL-CM6A.

Finally, the reduction in simulated plankton biomass in IPSL-CM5A (-6.7% for phytoplankton and -14.9% for zooplankton) is less extensive in IPSL-CM6A (-4.3% and -10.7% for phytoplankton and zooplankton, respectively; Table 3). It is interesting to note that despite opposite trends in projected NPP, both models simulate significant reductions in planktonic

biomass, and an associated trophic amplification, i.e. a larger decrease in zooplankton than phytoplankton (Lotze et al.,



2019; Kwiatkowski et al., 2019). This suggests that the potential impact on projections of upper trophic levels (e.g. Tittensor et al., 2018) should not differ as much as the differences in NPP projections between IPSL-CM5A and IPSL-CM6A indicate for ecosystem models forced by biomass changes. It also implies that changes in NPP may not be a robust proxy for diagnosing the potential impact of anthropogenic climate change on marine ecosystems.

**4 Conclusions and Recommendations**

Two versions of the IPSL ESM (IPSL-CM5A and IPSL-CM6A) are shown to project diverging global NPP trends in the 21st century under a similar high-emission scenario, because of the specificities of the diazotrophy parameterization employed in the different versions of the marine biogeochemical component PISCES. The use of additionnal PISCES versions confirms the role of diazotrophy parameterizations in driving divergent NPP responses in all subtropical gyres, with increased

(decreased) diazotrophy leading to increased (decreased) NPP. We identify both the thermal response and the treatment of stoichiometric N:P ratios to be of  importance for the future evolution of N-fixation in the future ocean. None of the of the PISCES versions used here perform significantly better when compared to observations or data-based reconstructions of surface nutrients, NPP and N-fixation rates. Under the same physical forcing, the divergent responses of N-fixation lead to significantly different deoxygenation and changes in organic matter export at the end of the 21st century, with larger

deoxygenation and weaker reduction in organic carbon export in the model version simulating a large warming-driven increase in N-fixation.  Despite these divergent NPP responses, all PISCES versions simulate decreasing trends of phytoplankton and zooplankton biomasses in the coming decades.

Although this study focuses on simulations performed with several versions of the same climate and marine biogeochemical models, similar conclusions have been drawn from using a series of ESMs that participated to CMIP5. Using an approach

combining 6 CMIP5 models and proxies for historical trends of N-fixation from the subtropical Pacific, Riche and Christian (2017) conclude that the environmental controls on ocean N-fixation remain elusive and that future trends are therefore highly uncertain. In addition, and using 9 CMIP5 ESMs, Wrightson and Tagliabue (2020) demonstrate that the future evolution of nitrogen fixation is a key determinant of the future trends in NPP in the tropics. Similar analysis using the new set of CMIP6 ESMs remains to be carried out.

Ultimately, the results of this study argue for a more robust treatment of marine nitrogen fixation in ESMs used for climate projections of ecosystem services.  This would suggest the need to include explicit diazotroph planktonic function groups in ESMs and to understand how differences between different nitrogen fixing groups affect projections. For instance, the main open-ocean autotrophic diazotrophs *Trichodesmium* and *Croccosphaera* that contribute around half of total nitrogen fixation (Zehr and Capone, 2020) show variability in their thermal performance curves (Fu et al., 2014), nutrient requirements (Saito

et al.. 2011) and even in their response to changing ocean pCO2 levels (Hutchins et al., 2013). Intriguingly, it also appears that temperature may play a fundamental role, with the resource costs of nitrogen fixation strongly dependant on temperature for both groups, which implies potential increases in future N-fixation rates may arise (Yang et al., 2021). We currently lack



an integrated assessment of how important these factors are in the context of a changing climate and the potential feedbacks onto NPP alterations.

Given the significant differences in the projections of N-fixation over the 21st century that are reported here or arise from model intercomparison studies (Riche and Christian, 2018; Wrightson and Tagliabue, 2021), it is a priority to derive methods to better constrain these projections. The paucity of direct N-fixation rate measurement in the present ocean (Landolfi et al., 2018), is limiting the development of such constraints and it is a priority to continue collecting such high-quality data to be able to extract significant temporal trends of N-fixation to be compared to model output. Other approaches, such as the use

of nitrogen isotopes from water column (Buchanan et al., 2021) or marine sediment cores to reconstruct past trends in N-fixation in contrasted regions of the world ocean (eg. Sherwood et al., 2014) will certainly offer new opportunities to constraint the modelled projections.

### Acknowledgments

The CMIP6 project at IPSL used the HPC resources of TGCC under the allocations 2016-A0030107732, 2017-
R0040110492, and 2018-R0040110492 (project gencmip6) provided by GENCI (Grand Équipement National de Calcul Intensif). This study benefited from the ESPRI (Ensemble de Services Pour la Recherche l'IPSL) computing and data center (https://mesocentre.ipsl.fr) which is supported by CNRS, Sorbonne Université, École Polytechnique, and CNES and through national and international grants. All authors acknowledge support from the French ANR project CIGOEF (grant ANR-17-CE32-0008-01). The authors also acknowledge support from the European Union's Horizon 2020 research and innovation

programs CRESCENDO (grant agreement No 641816), COMFORT (grant agreement No 820989), 4C (grant agreement No 821003) and ESM2025 (grant agreement No 101003536). LB received funding from the Chaire ENS-Chanel. AT received funding from the European Research Council (ERC) under the European Union's Horizon 2020 research and innovation programme (grant agreement no. 724289).

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
