# Peer review of "Diazotrophy as a key driver of the response of marine net primary productivity to climate change"

_Biogeosciences, 2021_

## Author Comment (AC1)

We would like to thank the 2 reviewers of our manuscript for their detailed and careful review. We respond to their comments hereafter point by point (in bold) and indicate what will be changed/updated in our revised manuscript.

**Responses to Referee #1 :**

This manuscript assesses simulations with five different global ocean biogeochemical models to assess uncertainties about the role that biological dinitrogen (N2) fixation will play in changes in ocean productivity and biogeochemistry in a warming climate. Generally, the analysis is sound, but the presentation is weak in places. I offer some suggestions below for ways to revise the paper to make the overall conclusions more compelling.

**>> We thank Referee #1 for his/her comments and suggestions for improving our manuscript.**

(1a) I think the comparison between IPSL-CM5A-LR and the offline simulations is presented in a misleading way. This is not an apples-to-apples comparison. An offline simulation will never reproduce the parent model exactly. Therefore the suite should really include PISCES-v1 run in the offline mode. Possibly the differences from the ESM would be small. But it would be nice if the reader could verify that. Consider cumulative CO2 uptake (Table 3). It differs by only 16 PgC between the two ESMs despite the massive increase in N2 fixation in CM6A (and different atmosphere and ocean models, and different emission scenarios). But all 3 offline simulations differ from CM5A by 50-90 PgC, despite the 'identical' (328) circulation. I suspect this has more to do with the inline/offline configuration than with the bgc model structure. If the authors want to test this, I think they could achieve this with an offline PISCES-v1 experiment.

**>> Indeed, the reviewer is right in suggesting that our offline approach does not reproduce the exact solution as in the online mode. The offline approach used here relies on monthly physical output (oceanic currents, temperature, salinity, diffusion coefficients, …) to force the biogeochemical model. These physical monthly output are then interpolated in time at the biogeochemical time-step and hence do not take into account short time-scale processes (e.g. vertical mixing events) with implications for both our mean biogeochemical state and its evolution. This is exemplified in the analyzed carbon cumulative uptake (Section 3.5) where differences between the different model versions are better explained by differences arising from the online/offline strategy than by the different versions of our biogeochemical model.**

**These considerations will be added to the revised manuscript (in the Methods section when describing the offline strategy, and in the Results section when the offline/online differences contribute to model differences, such as for the cumulative carbon uptake).**

**We are in the process of running an additional simulation complementing the suite of existing simulations with an offline PISCES-v1 experiment as suggested by the referee. If available in due course, this additional experiment will be integrated into our analysis to disentangle the role of online/offline vs. the impact of different biogeochemical schemes.**

(1b) I find some of the text explaining the results shown in Figure 5 vague or misleading. Some of the results are clearly robust to the differences in the physical ocean environment. For example, when we compare offline PISCES-v2 to CM6A the change in total N2 fixation is very similar (Figure 5b). But the net change in NPP differs by almost a factor of 2 (Figure 5a), and I think this glossed over in the text (316-321). The explanations of the differences among the offline models are also vague in places (e.g., 306-309).

**>> As described above, the different strategy between offline and online simulations partly explains the differences between the model versions. This will be clarified in the revised manuscript (methods and results sections).**

(2a) I don't think the analysis of mechanisms underlying the differences among models is as well presented as it could be. On 341-343 the text seems to be saying that N2 fixation declines due to warming in IPSL-CM5A-LR, which should not be the case, and is unaffected by DIN-inhibition. But the latter seems to be contradicted in the very next paragraph (348) and by the data in Figure 6. In the ESM L_N only goes down to ~0.8. But in the other models it doesn't change at all. So if 348 isn't referring to the ESM, what is it referring to?

**>> The referee is right that a clearer explanation of the mechanisms driving changes in N-fixation is necessary, particularly a better identification of when warming and/or DIN-inhibition is the dominant factor.**

**The method section was not clear enough in detailing the way N/P ratios are treated in the constant**

**(Redfield) ratio models (PISCES-v1, PISCES-v2, PISCES-v2fix). In the submitted draft, our text says:**

**"Finally, due to the fixed stoichiometric ratios between carbon, nitrogen and phosphorus in all organic components it is assumed that N-fixation is accompanied by a release of inorganic phosphorus to account for the fact that diazotrophy-derived organic matter is much richer in N than the standard Redfield assumptions in the model. For every mole of $N_2$ fixed by diazotrophy and instantaneously transferred into the ammonium pool, an additional 0.04 moles of phosphorus is added in the phosphate pool to represent the subsequent remineralization of the diazotrophy derived organic matter with an N:P ratio of 46:1 (Figure 2a and b). This additional P-source can be interpreted as deriving from the use of an unresolved dissolved organic phosphorus pool by diazotrophs".**

**As explained above, in all the models based on constant stoichiometric ratios (PISCES-v1, PISCES-v2 and PISCES-v2fix), we introduced some implicit parameterizations to take into account the high N/P ratios of diazotrophic organisms and their use of dissolved organic phosphorus (DOP) to cope with P-deficient environment. In the quota version of PISCES (PISCES-quota), both processes can be accounted for explicitly.**

**In PISCES-v1, we focused, when developing the model, on the high N/P ratios of diazotrophy. As the model is Redfieldian, we assumed that diazotrophic cyanobacteria take up PO4 with a mean N/P ratio equal to 1/46 (based on the literature) and release inorganic P (and ammonia) as a result of exudation and mortality with the typical Redfield ratio of 1/16. The difference is supposed to come from an unresolved labile DOP pool. As a consequence, a net release of about 0.04 moles of phosphorus is added for every mole of fixed N, as described above.**

**In PISCES-v2 and PISCES-v2fix, we updated this parameterization and instead focused on the unresolved source of labile DOP. In strongly P-limited areas, diazotrophic cyanobacteria have been found to use DOP as a source of P as do other phytoplankton groups (Cotner et al, 1992 ; Paytan and McLaughlin, 2007). We thus introduced a parameterization to mimic this source of P, which does depend on DOM levels and is inhibited when dissolved inorganic P is not limiting. This source does not depend anymore on the rate of N fixation as was the case in PISCES-v1.**

**This key differences between PISCES-v1 and PISCES-v2 / PISCES-v2fix versions will be detailed in the method section and will be made clearer on Figure 2. A short appendix, with the detailed equations and a justification for the choices that are made in the different PISCES versions, will be added to the revised manuscript.**

**Lastly, this clarification in the method section will enable a clearer description, in Section 3.3, of the mechanisms explaining the divergent response of N-fixation in PISCES-v1 vs. PISCES-v2.**

**In IPSL-CM5 (with PISCES-v1), warming first increases N-fixation, and this results in N-addition (through N-fixation) at the expense of P. In the oligotrophic subtropical gyres, surface waters then transit into P-limitation for phytoplankton (with N* becoming positive), which leads to the accumulation of inorganic nitrogen. N-fixation is hence limited by the increase of inorganic nitrogen concentrations and decreases.**

**In IPSL-CM6 (with PISCES-v2), warming also increases N-fixation due to the same parameterization of thermal sensitivity. But here, sustained low addition of phosphate (accounting for the implicit DOP remineralization and not depending on N-fixation rates) prevents any shift to P-limitation. N* stays negative, Inorganic nitrogen does not accumulate and N-fixation continues to increase.**

**In PISCES-v2fix and PISCES-quota, there is no transition to P-limitation as in PISCES-v2. In PISCES-v2fix, this is due to the same reason as in PISCES-v2. In PISCES-quota, this results from the preferential remineralization of DOP over DON and DOC. However, the revised thermal sensitivity inhibits N fixation above 26°C preventing any substantial increase with warming unlike what is simulated in PISCES-v2.**

**The paragraph from line 336 to line 443 will be rewritten and expanded.**

Similarly, in Section 3.5 I think there are several assertions that are questionable and not really supported with data. It isn't obvious to me why we would have a >2X larger decline in subsurface O2 in IPSL-CM6A-LR than IPSL-CM5A-LR, when export production is about the same and shows a very small (and negative) trend in IPSL-CM6ALR (Table 3). How does an increase in N2 fixation result in subsurface O2 depletion without affecting export? Possibly via DOM, but that is not substantiated or even discussed. And why would we assume that it is due to remineralization and not to circulation given the different climate models used? As in 3.3, I find the presentation here a bit careless.

**>> The impact of the different trends in N-fixation and NPP on sub-surface oxygen and export production is**

only briefly discussed in Section 3.5. These impacts will be discussed more clearly in the revised manuscript, stating that (1) the main factors driving sub-surface O2 depletion are indeed linked to ocean warming, ocean circulation changes and ocean stratification, and that (2) changes in NPP and export play a secondary role in modulating ocean deoxygenation. Indeed, the 2 model versions with little (positive or negative) changes in export production are the ones where ocean deoxygenation is the more severe, whereas the 3 model versions with significant decreases in export (IPSL-CM5A, PISCES-v2fix and PISCES-quota) are the ones where ocean deoxygenation is the weakest, because it is partially damped by decreasing O2-demand at the sub-surface. We provide here a simple plot showing how relative changes in export production relate to relative changes in sub-surface deoxygenation in our suite of model versions. This relationship will be further discussed in the revised manuscript.

[Figure]

Figure : Relative changes in Export production (at 100m) and in sub-surface oxygen (averaged between 100m and 600m) in 2080-2099 relative to 1986-2005.

I also do not think that the Western Pacific box in Figure 6a is as representative as is implied. If we take the North Pacific subtropical gyre as whole, the southwestern corner is probably the warmest part. At HOT, for example, annual mean SST is around 25C. So with the Breitbarth temperature function N2 fixation would initially increase, although it would decline if net warming exceeds ~2°C. In the Western Pacific box SST exceeds 26°C right from the beginning, so the decline is monotonic (Figure 6cd).

>> Our choice of the Western Pacific box in Figure 6a was meant to provide an illustrative example of the contrasted behavior of our different model versions in a place where the N-fixation response to anthropogenic climate change diverges between our 2 ESM versions (IPSL-CM5A vs. IPSL-CM6A, see the red regions on Figure 6a). We agree with the reviewer that this Western Pacific region does not capture the non-linear response to temperature in the model versions using the Breitbarth formulation (PISCES-v2fix, PISCES-quota) that would be seen elsewhere. In the revised version of the manuscript, we plan to provide additional figures in the supplementary materials showing the same analysis (as in Figure 6) but in other regions (e.g. in the Eastern N. Pacific subtropical gyre, or in the N. Atlantic subtropical gyre).

I also don't think including N* in Figure 6 is useful. N* is never mentioned up to this point (e.g., the Methods do not mention any of the models as having a dependence of N2 fixation on N*). Anyway, why would N* be negative at the surface in a region where a lot of N2 fixation occurs? Possibly this could be useful for evaluating the realism of the models wrt N/P stoichiometry, but at present it adds little to the analysis.

>> We agree with the reviewer that including N* is probably not needed to convey the main message of Figure 6, i.e. a shift from mostly N-limitation to P-limitation in IPSL-CM5, which leads to increasing levels of NO3 and DIN-inhibition of N-fixation. The panel showing N* trends will be removed from Figure 6 and the text from line 336 to line 443 will be rewritten and expanded.

In response to the question of negative N* and the occurrence of N fixation, we would like to point out that there is no incompatibility - N fixation is triggered (in the model) by low nitrate concentrations and phosphate availability, thus favored in negative N* regions. Despite the high rates of N fixation, nitrate is consumed very quickly by phytoplankton, keeping surface concentrations at very low levels.

(2b) Similarly, I'm not sure including N* in Figure 7 and Table 2 is a good idea. When I look at the observational distribution I think I understand why negative values occur in nutrient-depleted surface waters, but this is probably an artefact of the kind of data used. In the subtropical Atlantic, for example, Gruber and Sarmiento (10.1029/97GB00077) calculated that N* is positive in the subsurface waters and therefore that there was

probably net N2 fixation in the overlying surface waters. But in this plot we see a broad, contiguous area of negative N*. I think this is an artefact for two reasons. (1) In nutrient depleted surface waters almost all the nutrients are recycled. Therefore much of the DIN may be NH4, which is rarely measured. But P has no such redox chemistry: recycled PO4 and 'new' PO4 are the same, analytically. So using the gridded NO3+NO2 data product underestimates DIN and creates an artificially low N/P. (2) Concentrations are often lower than the analytical detection limit (ADL) for standard methods. Because the ADL for PO4 is not 16X lower than that for NO3, this again creates an artificially low DIN/DIP ratio. Why this also happens in the model is interesting. Do the model N* estimates include NH4? (Also, why are the columns ordered differently in Figure 7 and Table 2?)

**>> Evaluating our different model versions with "observed" N* is indeed not straightforward, many questions arise on the use of observational datasets to create the "data-based" N* map. In the revised manuscript, we will remove N* and include a direct comparison with surface nitrate and phosphate concentrations.**

(3) I general I do not think that the cluster analysis is adequately explained. Figure 4 is quite a lot of information to digest, and the presentation could be improved. Firstly, I would suggest that Figure 4(c) be moved to the top, as it contains the definitions of the colours. I think the black "n.c." segment on the colour bar should be removed (the only black on the map is over land), and the threshold for no change should be stated. The text states that cluster 4 – pink indicates that "growth rates increase without any significant change in N-fixation" (264-265) but does not state what is the criterion for a change to be considered significant. For panel (a) I would consider (1) using common ranges for the x and y axes, (2) stating in the caption that the greyscale applies (equally?) to all colours, and (3) making the 'no data' squares white instead of black. If using a common x and y range would interfere with the visual presentation, consider including it as an additional Supplementary version. I would change the caption to Figure 4(c) to something like "Global maps of the distribution of the five clusters". (Note that in my copy there is abcd in the caption, but no actual labels on the subplots. Also the labels in the figure all say mu but the caption says m; this might just be due to PDF rendition.) Conceptually, the description of the cluster analysis methodology is not very clear or specific. The only literature reference is to a 771 page textbook from 1991. So presumably this is a well-established methodology, but it will be unfamiliar to many readers, and I do not think the explanation given in the text is very illuminating. Nor can the reader easily trace it back to its cited source. Does P represent a probability? Probability of what? That X or Y will fall within a specific increment within (-1,1)? That it will be positive, negative, or NSDZ? What do X and Y represent? Normalized anomalies at the individual model grid points? Remapped to a regular grid? If it is point-by-point on whatever grid (Figure 4c+d), what defines a probability? The anomaly at each point has a unique value; sampling over some range of inputs is required to generate a probability distribution. Sorry I'm just not following exactly what was done here. A few sentences of explanation can go a long way in helping the reader to understand what is being presented.

**>> We agree with the reviewer and will improve the description and interpretation of the cluster analysis. Figure 4 will be changed following the reviewer's suggestions (order of panels, change of colors, more detailed caption). Accordingly, the method description will be expanded to better explain the cluster analysis used here.**

(4) I find Section 3.4 a bit disjointed. It almost feels like it is two separate sections spliced together and might better be split in two. 375-379 is like a wrapping-up of one topic and then a whole new one is introduced. It might also be a good idea to combine the paragraphs on 394-398 and 405-410 into one, so that the general background on emergent constraints leads directly to the application that is directly relevant here.

**>> We agree with the reviewer and will combine better the 2 ideas developed in this section – with a common introduction on the use of global datasets to evaluate our different model versions and/or to constrain projections using emergent constraint techniques.**

Methods
In 2.1 I would add a few sentences about the setup of the physical ocean models. On 126 it is stated that NEMO consists of ocean dynamics, sea ice and biogeochemistry. But there are many options for various physical process parameterizations and some configuration of these was used in the 'frozen' versions used for CMIP5/6. It would be good idea to state something in the Methods about how advection and mixing are done in the offline simulations.

**>> Agreed. We will expand the model description and refer to Aumont et al. (2015) for a better description of the offline configuration, and to Dufresne et al. (2013) and Boucher et al. (2020) for a detailed description of the ocean configuration used in the ESMs.**

The description of the modified N/P stoichiometry is a bit confusing. On 165-168 it is stated that using a high N/P for organic matter derived from diazotrophy is a change in PISCES-quota from the base case. But on 156 is says that PISCES-v2 has the same N/P=46 for organic matter derived from diazotrophy. It appears that the only difference is that in the base case the organic matter is implicit.

**>> We will add an appendix in which the description of the handling of N/P ratios for diazotrophy will be better explained.**

Some specifics:
26 change "model" to "models"

**>> Changed accordingly.**

56 CESM2s is an official model name?

**>> CESM2s changed to "CESM2 and CESM2-WACCM".**

65 delete "atmospheric" (see also 94)

**>> Changed accordingly.**

97 "unconstrained" misspelled (see also 359, 502)

**>> Changed accordingly.**

160 change "advanced" to "modified"

**>> Changed accordingly.**

206 "phytoplankton-realised growth" do not hyphenate

**>> Changed accordingly.**

253 change "ocean physics" to "physical ocean"

**>> Changed accordingly.**

267, 269 change "independent from" to "independent of"

**>> Changed accordingly.**

270 '' the reinforced relationship" Odd choice of words. How about: "confirms the intensified effect of diazotrophy on phytoplankton growth ..."?

**>> Text changed following the reviewer's suggestion.**

280 change "oceanic" to "ocean" (actually I think all 4 occurrences of "oceanic" could be changed to "ocean", but this one in particular)

**>> Changed accordingly.**

297 "The increase in N-fixation is dampened everywhere as compared to PISCES-v2, with even regions where N-fixation decreases in PISCES-v2fix" How about: "The increase in Nfixation is small everywhere compared to PISCES-v2; there are even regions where Nfixation decreases in PISCES-v2fix"?

**>> Text changed following the reviewer's suggestion.**

303 change "only slightly increases" to " increases only slightly"

**>> Changed accordingly.**

362 "better performance scores" I would not use this term unless the 'score' is defined somewhere.

**>> The score that the sentence is referring is based on the use of RMSE. This will be better stated in the revised manuscript.**

467 change "under a similar high-emission scenario" to "under similar high-emission scenarios"

**>> Changed accordingly.**

468 "additional" misspelled

**>> Changed accordingly.**

471 "of the of the"

**>> Changed accordingly.**

479 change "participated to" to "participated in"

**>> Changed accordingly.**

481/496 Riche and Christian: 2017 or 2018? (see also 98)

**>> Reference to Riche and Christian will be corrected. The exact reference is:**
**Riche, OGJ and Christian, JR 2018 Ocean dinitrogen fixation and its potential effects on ocean primary production in Earth system model simulations of anthropogenic warming. Elem Sci Anth, 6: 16. DOI: https://doi.org/10.1525/elementa.277**

485-494 It might be a good idea to mention Pahlow et al and Inomura et al in this paragraph, as they were described on 423 as presenting "more mechanistic models of N fixation".

**>> References to Pahlow et al. and Inomura et al. will be added to this paragraph.**

Figure 7 is too low resolution. When I expand it to be readable, the fonts are blurry.

**>> Figure 7 will be provided at higher resolution.**
* * *
**Responses to Referee #2:**

This study by Bopp et al. investigates how changes to N2 fixation affects NPP in a suite of climate model simulations (IPSL) with PISCES biogeochemistry under historical and future warming scenarios. They simulate 5 different model versions, 4 of which apply different formulations for biogeochemistry, most notably N2 fixation, and 1 simulation with different ocean physical circulation and resolution. The simulation with higher resolution (IPSLCM6A) did not significantly affect N2 fixation compared to the simulation with similar biogeochemistry and lower resolution (PISCES-v2) so they mainly focused this study on the differences in N2 fixation parameterizations.

They implemented a variety of different N2 fixation parameterizations including phosphorus limitation, temperature-dependent growth, elemental stoichiometry, and underlying biogeochemistry (the latter not well described in the paper). Their model simulation with a strong increase in N2 fixation (PISCES-v2) caused a strong increase in NPP, whereas the other simulations with a slight increase or decrease in N2 fixation projected a decrease in NPP throughout the 21st century. They focus on a region in the western tropical North Pacific to better understand the mechanisms responsible for the difference in these simulations. The one clear driver for driving the high N2 fixation rates in PISCES-v2 was the exponential temperature-dependence growth rate, whereas the PISCES-quota simulations using the Breitbarth et al. 2007 bell-shape reduced diazotroph growth rate at high >26°C temperatures.

Overall I think this is a fine manuscript that highlights the importance of N2 fixation on NPP, which is often neglected or not analyzed in model simulations. Another aspect I liked was how the authors distinguished between NPP and phytoplankton biomass instead of only focusing on NPP like most studies.

**>> We thank Referee #2 for his positive comments on our manuscript.**

My one major issue with the manuscript is the insufficient description of which processes are causing N vs. P vs. Fe limitation and thus driving the N2 fixation response in the different biogeochemistry model simulations, but this can be addressed in revisions. This information is included in some of the references, but given the emphasis on N2 fixation in the paper I think some additional details should be included and discussed in this manuscript as well.

**>> We agree with Reviewer #2 that a more detailed description of the different PISCES versions used here is needed in our manuscript. We will (1) expand the description of the different model versions in the Methods section, and (2) include an appendix with the detailed equations used for N-fixation.**

Major Comment
The one critical issue I do not understand in this study is why N2 fixation in IPSL-CM5A-LR does not increase more similarly to IPSL-CM6A-LR. Since they have the same temperature dependent growth rate, the authors state this is "due to the Ln term (limitation of excess inorganic nitrogen)" (line 339). However, all N2 fixation parameterization contain this Ln term (Figure 2). Why does Ln only decrease in IPSL-CM5A-LR and not any of the other simulations? My guess is that it is caused by some changes in the underlying biogeochemistry (e.g. iron limitation or denitrification), which is only mentioned in one brief sentence in the description section 2.2 (lines 143-145) and not further discussed. The processes that contribute to the decreasing Ln term in IPSL-CM5A-LR but not the other simulations should be described and discussed to understand these results presented here.

**>> Again, we agree with Reviewer #2 (Reviewer #1 has a similar comment). We reproduce here the response to Reviewer #1.**

**The method section was not clear enough in detailing the way N/P ratios are treated in the constant (Redfield) ratio models (PISCES-v1, PISCES-v2, PISCES-v2fix). In the submitted draft, our text says:**

**"Finally, due to the fixed stoichiometric ratios between carbon, nitrogen and phosphorus in all organic components it is assumed that N-fixation is accompanied by a release of inorganic phosphorus to account for the fact that diazotrophy-derived organic matter is much richer in N than the standard Redfield assumptions in the model. For every mole of N₂ fixed by diazotrophy and instantaneously transferred into the ammonium pool, an additional 0.04 moles of phosphorus is added in the phosphate pool to represent the subsequent remineralization of the diazotrophy derived organic matter with an N:P ratio of 46:1 (Figure 2a and b). This additional P-source can be interpreted as deriving from the use of an unresolved dissolved organic phosphorus pool by diazotrophs".**

**In all the models based on constant stoichiometric ratios (PISCES-v1, PISCES-v2 and PISCES-v2fix), we introduced some implicit parameterizations to take into account the high N/P ratios of diazotrophic organisms and their use of dissolved organic phosphorus (DOP) to cope with P-deficient environment. In the quota version of PISCES (PISCES-quota), both processes can be accounted for explicitly.**

**In PISCES-v1, we focused, when developing the model, on the high N/P ratios. As the model is Redfieldian, we assumed that diazotrophic cyanobacteria take up PO4 with a mean N/P ratio equal to 1/46 (based on the literature) and release inorganic P (and ammonia) as a result of exudation and mortality with the typical Redfield ratio of 1/16. The difference is supposed to come from an unresolved labile DOP pool. As a consequence, a net release of about 0.04 moles of phosphorus is added for every mole of fixed N, as described above.**

**In PISCES-v2 and PISCES-v2fix, we updated this parameterization and instead focused on the unresolved source of labile DOP. In strongly P-limited areas, diazotrophic cyanobacteria have been found to use DOP as a source of P as do other phytoplankton groups (Cotner et al, 1992 ; Paytan and McLaughlin, 2007). We thus introduced a parameterization to mimic this source of P, which does depend on DOM levels and is inhibited when dissolved inorganic P is not limiting. This source does not depend anymore on the rate of N fixation as was the case in PISCES-v1.**

**This key differences between PISCES-v1 and PISCES-v2 / PISCES-v2fix versions will be detailed in the method section and will be made clearer on Figure 2. A short appendix, with the detailed equations and a justification for the choices that are made in the different PISCES versions, will be added to the revised manuscript.**

**Lastly, this clarification in the method section will enable a clearer description, in Section 3.3, of the mechanisms explaining the divergent response of N-fixation in PISCES-v1 vs. PISCES-v2.**

**In IPSL-CM5 (with PISCES-v1), warming first increases N-fixation, and this results in N-addition (through N-**

fixation) at the expense of P. In the oligotrophic subtropical gyres, surface waters then transit into P-limitation for phytoplankton (with N* becoming positive), which leads to the accumulation of inorganic nitrogen. N-fixation is hence limited by the increase of inorganic nitrogen concentrations and decreases.

In IPSL-CM6 (with PISCES-v2), warming also increases N-fixation due to the same parameterization of thermal sensitivity. But here, sustained low addition of phosphate (accounting for the implicit DOP remineralization and not depending on N-fixation rates) prevents any shift to P-limitation. N* stays negative, Inorganic nitrogen does not accumulate and N-fixation continues to increase.

In PISCES-v2fix and PISCES-quota, there is no transition to P-limitation as in PISCES-v2. In PISCES-v2fix, this is due to the same reason as in PISCES-v2. In PISCES-quota, this results from the preferential remineralization of DOP over DON and DOC. However, the revised thermal sensitivity inhibits N fixation above 26°C preventing any substantial increase with warming unlike what is simulated in PISCES-v2.

The paragraph from line 336 to line 443 will be rewritten and expanded.

Minor Comments :
Lines 144-145: "… external sources of nutrients and the treatment of water-sediment interactions" Following my comment above, these processes can significantly impact N2 fixation rates. While I understand they are described in previous papers, the important processes controlling N2 fixation in these simulations presented here should be described and discussed.

>> We agree with the reviewer and will include more details on the external sources of nutrient (rivers, atmospheric deposition, sediment mobilization) as well as on nitrogen sink (e.g. denitrification).

Line 155: "… phosphorus is added to the phosphate pool" I thought diazotrophs are responsible for consuming PO4 and DOP to very low levels e.g (Mather et al., 2008), not providing a source. These fluxes are generally small among the entire circulation-biogeochemical system so it is likely not significant, but I am concerned that this PO4 source could be supporting additional NPP in an unrealistic way.

>> The reviewer is right in assuming diazotrophs are consuming PO4 and DOP. But due to the constant-ratio used for all organic pools in most model versions (PISCES-v1, PISCES-v2, PISCES-v2fix), we account for an unresolved source of phosphate derived from the remineralization of an implicit labile dissolved organic phosphorus pool. In PISCES-quota, this additional source is not required as DOP is explicitly modeled. The very similar N-fixation changes with climate change simulated by PISCES-v2fix and PISCES-quota suggest that our (crude) parameterization is acting in a consistent manner.

Line 158: "… annual restoring of global mean PO4 concentration …" I am surprised to see a PO4 restoring term in a prognostic climate model. I assume this is applied only during the spin-up and not historical/future projections? Does this occur at all depths?

>> The restoring of global mean PO4 concentration conserves the global phosphate pool – this is especially important for the long spin-up simulation of the coupled model so that the global P budget does not drift. This restoring term is applied everywhere (at all depths) and modifies phosphate concentration relatively, thus acting preferentially in the deep ocean where PO4 concentrations are higher. Overall, this term represents a restoring timescale of about 10,000 years.

The restoring term is retained for all simulations used here (historical/future projections included), but represents a very limited flux, which compensates for initial disequilibrium between P-input from rivers and atmospheric deposition (and implicit DOP remineralization), and P-loss through sediment burial.

Line 306: "The regional changes of NPP resemble those simulated in IPSL-CM5A" Is this also true for N2 fixation?

>> Yes. This is true also for N2 fixation – the text will be modified accordingly.

Lines 307-309: "… differences between PISCES-quota and two other offline simulations originate from the major developments in PISCES-quota such as variable C:N:P stoichiometry and the inclusion of a third phytoplankton …" This comes back to my major comment. It is interesting that newest (PISCES-quota) and the oldest formulation

presented (PISCES-v1 in IPSL-CM5A-LR) give very similar projections of changes to NPP and N2 fixation (Figure 5a,b), despite their large differences in biogeochemical model structure/parameterizations, whereas the in-between version (PISCES-v2) shows a significant difference. I would be interested to know which changes mainly contributed to the large increase in N2 fixation and NPP and PISCES-v2 compared to v1. This statement specifically mentions stoichiometry and the additional phytoplankton type, but I thought it was pretty clear that the change in temperature-dependent N2 fixation growth rate was responsible for these decreases in N2 fixation and NPP in PISCES-quota compared to v2 as described in the following section so I am confused by this statement.

**>> Section 3.3, describing the mechanisms explaining the divergent responses of N-fixation across our different model versions will be expanded and rewritten. As described above, the respective role of DIN-inhibition (limiting N-fixation in PISCES-v1) and warming (limiting N-fixation in PISCES-v2fix and PISCES-quota) will be made clearer.**

**It is indeed interesting to note that our 2 "extreme" model versions, the least complex (PISCES-v1) and the most complex (PISCES-quota) display quite similar trends for N-fixation and subsequently NPP. If we trust the more advanced version (explicit variable stoichiometric ratios for C:N:P, no need for an implicit DOP remineralization, inclusion of a third phytoplankton functional group) more than the others, this would suggest that the simulated Nfixation / NPP response in PISCES-v1 (within IPSL-CM5) is "correct" but for the wrong reasons. It also demonstrates the importance of thermal performance curves, with that used in PISCES-v2fix and PISCES-quota, (an optimal temperature of 26°C being questionable, Fu et al. 2014).**

Please describe how variable C:N:P stoichiometry and a third phytoplankton class causes higher N2 fixation because this is not clear. I think this emphasizes the need to better understand the key processes causing the N2 fixation differences in the different simulations.

**>> We agree with the reviewer and will provide, as detailed above, more material in the Methods section and a clearer description of the mechanisms driving the divergent N-fixation trends in the different model versions.**

Mather, R. L., Reynolds, S. E., Wolff, G. A., Williams, R. G., Torres-Valdes, S., Woodward, E. M. S., Landolfi, A., Pan, X., Sanders, R., and Achterberg, E. P.: Phosphorus cycling in the North and South Atlantic Ocean subtropical gyres, Nature Geoscience, 1, 439-443, 10.1038/ngeo232, 2008.

**References cited:**

**Cotner, J.B., Jr., and Wetzel, R.G. (1992). Uptake of dissolved inorganic and organic bphosphorus compounds by phytoplankton and bacterioplankton. Limnology and Oceanography *37*, 232–243.**

**Fu, F.-X., Yu, E., Garcia, N. S., Gale, J., Luo, Y., Webb, E. A., and Hutchins, D. A. (2014) Differing responses of marine N2 fixers to warming and consequences for future diazotroph community structure, 72, 33–46.**

**Paytan, A., and McLaughlin, K. (2007). The Oceanic Phosphorus Cycle. Chem. Rev. *107*, 563–576.**

---

## Author Response (AR1)

Dear Editor,

We would like to thank you for your positive comments on our manuscript and apologize for the time it took us to submit a revised manuscript; the delay was primarily due to the need to run an additional simulation with an "old" version of our biogeochemical model.

We followed most of the reviewers' comments in revising our manuscript, and we would like to thank both reviewers for their detailed and thoughtful comments that improved the manuscript. In particular, we have:

(1) included a new offline simulation (using PISCES-v1), so that the analysis of the mechanisms now relies primarily on a consistent set of simulations.

(2) improved the description of the nitrogen fixation parameterizations used in PISCES-v1 and PISCES-v2, identifying differences in how phosphorus is treated in these parameterizations, with implications for identifying the main mechanisms causing model discrepancies.

(3) removed the cluster analysis, which was not adequately explained and does not add much to the logic of the manuscript.

(4) expanded the mechanisms section to clarify the causes of differential responses of nitrogen fixation to climate change.

Note that almost all figures and tables have changed to account for the inclusion of the new simulation. In addition, we have modified the manuscript following all other comments from the 2 reviewers. Please find below a detailed point-by-point response to all reviewers' comments.

Best regards,

Laurent Bopp, on behalf of all co-authors.

\_\_\_\_

Responses to Referee #1 :

This manuscript assesses simulations with five different global ocean biogeochemical models to assess uncertainties about the role that biological dinitrogen (N2) fixation will play in changes in ocean productivity and biogeochemistry in a warming climate. Generally, the analysis is sound, but the presentation is weak in places. I offer some suggestions below for ways to revise the paper to make the overall conclusions more compelling.

**>> We thank Referee #1 for his/her comments and suggestions for improving our manuscript.**

(1a) I think the comparison between IPSL-CM5A-LR and the offline simulations is presented in a misleading way. This is not an apples-to-apples comparison. An offline simulation will never reproduce the parent model exactly. Therefore the suite should really include PISCES-v1 run in the offline mode. Possibly the differences from the ESM would be small. But it would be nice if the reader could verify that. Consider cumulative CO2 uptake (Table 3). It differs by only 16 PgC between the two ESMs despite the massive increase in N2 fixation in CM6A (and different atmosphere and ocean models, and different emission scenarios). But all 3 offline simulations differ from CM5A by 50-90 PgC, despite the 'identical' (328) circulation. I suspect this has more to do with the inline/offline configuration than with the bgc model structure. If the authors want to test this, I think they could achieve this with an offline PISCES-v1 experiment.

>> Indeed, the reviewer is right in suggesting that our offline approach does not reproduce the exact solution as in the online mode. The offline approach used here relies on monthly physical output (oceanic currents, temperature, salinity, diffusion coefficients, ...) to force the biogeochemical model. These physical monthly output are then interpolated in time at the biogeochemical time-step and hence do not take into account short time-scale processes (e.g. vertical mixing events) with implications for both our mean biogeochemical state and its evolution.

To overcome this limitation, we have performed and added a new offline simulation using the PISCES-v1 biogeochemical model forced by outputs of IPSL-CM5A-LR (as for the other offline versions). This enables to consistently test the offline/online approach and to rely on a set of consistent offline simulations to identify the mechanisms driving the N-fixation responses and discussing the implications.

**In the analysis of the offline simulations, we have added the following paragraph:**

The PISCES-v1 version (used in IPSL-CM5A) forced in offline mode with the IPSL-CM5A output over 1850-2100 gives broadly similar results than the ones we obtain with IPSL-CM5A, i.e a decrease in NPP and in N-fixation of -13.2% and -14.9%, in 2080-2099, respectively (Table 1, Figure 4.a and 4.b). Spatially, the NPP and N-fixation changes obtained with PISCES-v1 in offline mode strongly resemble those from IPSL-CM5A, as can be seen when comparing Figure 3 (c,e) to Figure 4 (c,d).

**All sections, figures and tables have been chaged accordingly to refer to this new simulation when appropriate (incl. Figure 3, Figure 4, Figure 5, Figure 6, Table 1, Table 2, Table 3).**

(1b) I find some of the text explaining the results shown in Figure 5 vague or misleading. Some of the results are clearly robust to the differences in the physical ocean environment. For example, when we compare offline PISCES-v2 to CM6A the change in total N2 fixation is very similar (Figure 5b). But the net change in NPP differs by almost a factor of 2 (Figure 5a), and I think this glossed over in the text (316-321). The explanations of the differences among the offline models are also vague in places (e.g., 306-309).

>> As described above, the different strategy between offline and online simulations partly explains the differences between the model versions. This is now clarified. The addition of a new offline version (see above) now enables a more consistent identification of the mechanisms linked to the different biogeochemical version – Section 3.3 on Mechanisms determining N-fixation responses now focuses solely on offline simulations (all forced with the same physical output).

(2a) I don't think the analysis of mechanisms underlying the differences among models is as well presented as it could be. On 341-343 the text seems to be saying that N2 fixation declines due to warming in IPSL-CM5A-LR, which should not be the case, and is unaffected by DIN-inhibition. But the latter seems to be contradicted in the very next paragraph (348) and by the data in Figure 6. In the ESM L\_N only goes down to ~0.8. But in the other models it doesn't change at all. So if 348 isn't referring to the ESM, what is it referring to?

>> The referee is right that a clearer explanation of the mechanisms driving changes in N-fixation is necessary, particularly a better identification of when warming and/or DIN-inhibition is the dominant factor.

We have now expanded the Method section and updated Figure 2 to better detail the way N/P ratios are treated in the constant (Redfield) ratio models (PISCES-v1, PISCES-v2, PISCES-v2fix). The new paragraph reads:

" In both PISCES versions, N-fixation is represented implicitly as a source of ammonium, i.e. without an explicit diazotroph plankton functional type (Figure 2). N-fixation is restricted to warm-waters (> 20°C) and increases exponentially with temperature following a Q-10 value of 1.9 as for all autotrophic processes in PISCES (Aumont and Bopp, 2006; Aumont et al., 2015). *N*-fixation is limited by the availability of light and iron, and favoured in low-nitrogen (NO3 and NH4) environments. PISCES-v1 and PISCES-v2 differ in their treatment of phosphorus limitation on N-fixation, which is absent in PISCES-v1 but combined with iron limitation in PISCES-v2. In PISCES-v1, due to the fixed stoichiometric ratios between carbon, nitrogen and phosphorus in all organic components it is assumed that N-fixation is accompanied by a release of inorganic phosphorus to account for the fact that diazotrophy-derived organic matter is much richer in N than the standard Redfield assumptions in the model. This additional P-source is interpreted as deriving from the use of an unresolved Dissolved Organic Phosphorus (DOP) pool by the implicit diazotrophs. Thus, in PISCES-v1, for every mole of  $N_2$ fixed by diazotrophy and instantaneously transferred into the ammonium pool, an additional 0.04 moles of phosphorus is added in the phosphate pool to represent the subsequent remineralization of the diazotrophy-derived organic matter with an N:P ratio of 46:1 (Figure 2a). In PISCES-v2, this parameterization has been changed and instead includes an unresolved source of inorganic phosphorus from labile DOP, independently of N-fixation. In strongly Plimited areas, diazotrophic cyanobacteria use DOP as a source of P, but this is the case also for other phytoplankton groups (Cotner et al, 1992; Paytan and McLaughlin, 2007). The parameterization thus mimics this source of P, which depends on simulated dissolved organic matter concentrations and is inhibited when dissolved inorganic P is not limiting phytoplankton growth (Figure 2b). In PISCES-v2, this P-source is therefore not dependent on the rate of Nfixation as it is in PISCES-v1".

**The section on Mechanisms determining the N-fixation response (Section 3.3) has also been expanded in order to clarify the role of warming vs. DIN-inhibition in driving the N-fixation response. The new paragraph now reads:**

"In PISCES-v2, the limitation terms due to light, phosphate, iron and excess nitrate remain inoperative and N-fixation responds almost exclusively to temperature (Figure 5b). Indeed,  $NO_3$  concentrations remain close to zero (with N\* values, defined as  $NO_3 - 16*PO_4$ , remaining slightly negative), and the  $L_N$  term has no influence, allowing N-fixation to increase in response to temperature. In PISCES-v1, on the other hand, the higher and increasing nitrate concentrations (Figure 5e), resulting in positive N\* values (Figure 5h), lead to a limitation and even to a decrease in N-fixation due to the  $L_N$  term (limitation by excess inorganic nitrogen, Figure 5f) (see Figure 2b, equation (2)).

The differential response of N-fixation in PISCES-v1 and PISCES-v2 is thus related to how the parameterization of N-fixation simulates inorganic N and P inputs to surface waters (See Methods 2.2). In PISCES-v1, surface waters in the oligotrophic subtropical gyres are P-limited for phytoplankton (N\* being positive). Warming first increases N fixation, resulting in a continuous addition of N (through N-fixation) at the expense of P. Nitrogen fixation is hence progressively limited by the accumulation of inorganic nitrogen and thus decreases. In PISCES-v2, warming also increases N-fixation due to the same parameterization of the thermal sensitivity of N-fixation. But in this version, the sustained low addition of phosphate (which accounts for the implicit remineralization of DOP and is independent of N-fixation rates) prevents any shift to P limitation. N\* remains negative, inorganic N does not accumulate, and N-fixation continues to increase

In PISCES-v2fix and PISCES-quota, NO3 concentrations remain close to zero as in PISCESv1 (Figure 5e) and the excess nitrogen limitation term has no influence, remaining close to 1 (Figure 5f). But contrary to PISCES-v1, N-fixation decreases slightly (Figure 5b) due to warming and the use of the bell-shape temperature sensitivity function (Figure 5d)".

Similarly, in Section 3.5 I think there are several assertions that are questionable and not really supported with data. It isn't obvious to me why we would have a >2X larger decline in subsurface O2 in IPSL-CM6A-LR than IPSL-CM5A-LR, when export production is about the same and shows a very small (and negative) trend in IPSL-CM6ALR (Table 3). How does an increase in N2 fixation result in subsurface O2 depletion without affecting export? Possibly via DOM, but that is not substantiated or even discussed. And why would we assume that it is due to remineralization and not to circulation given the different climate models used? As in 3.3, I find the presentation here a bit careless.

>> The impact of the different trends in N-fixation and NPP on sub-surface oxygen and export production was only briefly discussed in Section 3.5. We now state that the main factors driving sub-surface O2 depletion are indeed linked to ocean warming, ocean circulation changes and ocean stratification, and that changes in NPP and export do play a significant role in modulating ocean deoxygenation. Indeed, the 2 model versions with little (positive or negative) changes in export production are the ones where ocean deoxygenation is the more severe (IPSL-CM6A, PISCES-v2), whereas the 4 model versions with significant decreases in export (IPSL-CM5A, PISCES-v1, PISCESv2fix and PISCES-quota) are the ones where ocean deoxygenation is the weakest, because it is partially damped by decreasing O2-demand at the sub-surface. We provide here a simple plot showing how relative changes in export production relate to relative changes in sub-surface deoxygenation in our suite of model versions.

Figure : Relative changes in Export production (at 100m) and in sub-surface oxygen (averaged between 100m and 600m) in 2080-2099 relative to 1986-2005 for all model versions used in this study.

This relationship between relative changes in export and changes in sub-surface oxygen concentrations is now explicitly mentioned in our revised manuscript (Section 3.5).

I also do not think that the Western Pacific box in Figure 6a is as representative as is implied. If we take the North Pacific subtropical gyre as whole, the southwestern corner is probably the warmest part. At HOT, for example, annual mean SST is around 25C. So with the Breitbarth temperature function N2 fixation would initially increase, although it would decline if net warming exceeds ~2°C. In the Western Pacific box SST exceeds 26°C right from the beginning, so the decline is monotonic (Figure 6cd).

>> Our choice of the Western Pacific box in Figure 6a was meant to provide an illustrative example of the contrasted behavior of our different model versions in a place where the N-fixation response to anthropogenic climate change diverges between our 2 ESM versions (IPSL-CM5A vs. IPSL-CM6A, see the red regions on Figure 5a). We have now updated Figure 5 with the addition of the new PISCES-v1 offline simulation. We also now provide an additional supplementary figure, showing the same analysis but for the HOT region as suggested by the reviewer (Figure S1). The conclusions remain the same than when using our Western Pacific region, distinguishing the differential role of warming vs. DIN-inhibition.

I also don't think including N\* in Figure 6 is useful. N\* is never mentioned up to this point (e.g., the Methods do not mention any of the models as having a dependence of N2 fixation on N\*). Anyway, why would N\* be negative at the surface in a region where a lot of N2 fixation occurs? Possibly this could be useful for evaluating the realism of the models wrt N/P stoichiometry, but at present it adds little to the analysis.

>> We decided to keep N\* on Figure 5 (and supplementary Figure 1), as it enables to show in which model versions phytoplankton growth is more likely to be N-limited or P-limited. This diagnostic clearly differentiates PISCES-v2 (with positive N\* values, and P-limitation for phytoplankton) from all other PISCES versions (with negative N\* values).

In response to the question of negative N\* and the occurrence of N fixation, we would like to point out that there is no incompatibility - N fixation is triggered (in the model) by low nitrate concentrations and phosphate availability, thus favored in negative N\* regions. Despite the high rates of N fixation, nitrate is consumed very quickly by phytoplankton, keeping surface concentrations at very low levels.

(2b) Similarly, I'm not sure including N\* in Figure 7 and Table 2 is a good idea. When I look at the observational distribution I think I understand why negative values occur in nutrient-depleted surface waters, but this is probably an artefact of the kind of data used. In the subtropical Atlantic, for example, Gruber and Sarmiento (10.1029/97GB00077) calculated that N\* is positive in the subsurface waters and therefore that there was probably net N2 fixation in the overlying surface waters. But in this plot we see a broad, contiguous area of negative N\*. I think this is an artefact for two reasons. (1) In nutrient depleted surface waters almost all the nutrients are recycled. Therefore much of the DIN may be NH4, which is rarely measured. But P has no such redox chemistry: recycled PO4 and 'new' PO4 are the same, analytically. So using the gridded NO3+NO2 data product underestimates DIN and creates an artificially low N/P. (2) Concentrations are often lower than the analytical detection limit (ADL) for standard methods. Because the ADL for PO4 is not 16X lower than that for NO3, this again creates an artificially low DIN/DIP ratio. Why this also happens in the model is interesting. Do the model N\* estimates include NH4? (Also, why are the columns ordered differently in Figure 7 and Table 2?)

>> Evaluating our different model versions with "observed" N\* is indeed not straightforward, many questions arise on the use of observational datasets to create the "data-based" N\* map. In the revised manuscript, we have removed N\* and include a direct comparison with surface nitrate and phosphate concentrations.

(3) I general I do not think that the cluster analysis is adequately explained. Figure 4 is quite a lot of information to digest, and the presentation could be improved. Firstly, I would suggest that Figure 4(c) be moved to the top, as it contains the definitions of the colours. I think the black "n.c." segment on the colour bar should be removed (the only black on the map is over land), and the threshold for no change should be stated. The text states that cluster 4 – pink indicates that "growth rates increase without any significant change in N-fixation" (264-265) but does not state what is the criterion for a change to be considered significant. For panel (a) I would consider (1) using common ranges for the x and y axes, (2) stating in the caption that the greyscale applies (equally?) to all colours, and (3) making the 'no data' squares white instead of black. If using a common x and y range would interfere with the visual presentation, consider including it as an additional Supplementary version. I would change the caption to Figure 4(c) to something like "Global maps of the distribution of the five clusters". (Note that in my copy there is abcd in the caption, but no actual labels on the subplots. Also the labels in the figure all say mu but the caption says m; this might just be due to PDF rendition.) Conceptually, the description of the cluster analysis methodology is not very clear or specific. The only literature reference is to a 771 page textbook from 1991. So presumably this is a well-established methodology, but it will be unfamiliar to many readers, and I do not think the explanation given in the text is very illuminating. Nor can the reader easily trace it back to its cited source. Does P represent a probability? Probability of what? That X or Y will fall within a specific increment within (-1,1)? That it will be positive, negative, or NSDZ? What do X and Y represent? Normalized anomalies at the individual model grid points? Remapped to a regular grid? If it is point-by-point on whatever grid (Figure 4c+d), what defines a probability? The anomaly at each point has a unique value; sampling over some range of inputs is required to generate a probability distribution. Sorry I'm just not following exactly what was done here. A few sentences of explanation can go a long way in helping the reader to understand what is being presented.

>> We agree with the reviewer and have decided to remove the cluster analysis from the manuscript. The method is not straightforward to explain and does not bring much more information that a simple comparison of the N-fixation / NPP changes maps as detailed in Section 3.1 and 3.2. The corresponding Methods and Results paragraphs as well as the initial Figure 4 have been removed in the revised draft.

(4) I find Section 3.4 a bit disjointed. It almost feels like it is two separate sections spliced together and might better be split in two. 375-379 is like a wrapping-up of one topic and then a whole new one is introduced. It might also be a good idea to combine the paragraphs on 394-398 and 405-410 into one, so that the general background on emergent constraints leads directly to the application that is directly relevant here.

>> We agree with the reviewer that Section 3.4 was a bit disjointed. We have changed Section 3.4 so that the 2 topics are better connected, and merged the paragraphs on lines 394-398 and 405-401 as suggested by the reviewer.

**Methods**

In 2.1 I would add a few sentences about the setup of the physical ocean models. On 126 it is stated that NEMO consists of ocean dynamics, sea ice and biogeochemistry. But there are many options for various physical process parameterizations and some configuration of these was used in the 'frozen' versions used for CMIP5/6. It would be good idea to state something in the Methods about how advection and mixing are done in the offline simulations.

**>> Agreed. The description of the physical NEMO configuration has been expanded to better describe some imporvements that have been incorporated in IPSL-CM from its 5th to its 6th version. The text now reads:**

"The NEMO ocean model comprises 3 components, i.e. ocean dynamics (NEMO-OPA), seaice dynamics and thermodynamics (NEMO-LIM) and marine biogeochemistry (NEMO-PISCES). All of these ocean components have been updated from IPSL-CM5A-LR to IPSL-CM6A-LR, from version 3.2 to version 3.6 of NEMO (Madec et al., 2017; Rousset et al., 2015; Aumont et al., 2015), with the addition of a nonlinear free surface, a parameterization of mixing in the mixed layer due to submescale processes, and an energy constrained parameterization of mixing due to internal tides for NEMO-OPA, and a new multicategory halothermodynamic sea ice model for NEMO-LIM. The changes in the marine biogeochemistry component (NEMO-PISCES) are described below".

The description of the modified N/P stoichiometry is a bit confusing. On 165-168 it is stated that using a high N/P for organic matter derived from diazotrophy is a change in PISCESquota from the base case. But on 156 is says that PISCES-v2 has the same N/P=46 for organic matter derived from diazotrophy. It appears that the only difference is that in the base case the organic matter is implicit.

>> The description of the modified N/P stoichiometry for the different model versions has been entirely re-written. The paragraph now provides more details on how diazotrophy is represented in the different model versions. In particular, we now detail how the addition of phosphorus in the "redfieldien" PISCES versions (PISCES-v1, PISCES-v2, PISCES-v2fix) is treated differently.

**The text now reads:**

"In both PISCES versions, N-fixation is represented implicitly as a source of ammonium, i.e. without an explicit diazotroph plankton functional type (Figure 2). N-fixation is restricted to warm-waters (> 20°C) and increases exponentially with temperature following a Q-10 value of 1.9 as for all autotrophic processes in PISCES (Aumont and Bopp, 2006; Aumont et al., 2015). *N*-fixation is limited by the availability of light and iron, and favoured in low-nitrogen (NO3 and NH4) environments. PISCES-v1 and PISCES-v2 differ in their treatment of phosphorus limitation on N-fixation, which is absent in PISCES-v1 but combined with iron limitation in PISCES-v2. In PISCES-v1, due to the fixed stoichiometric ratios between carbon, nitrogen and phosphorus in all organic components it is assumed that N-fixation is accompanied by a release of inorganic phosphorus to account for the fact that diazotrophy-derived organic matter is much richer in N than the standard Redfield assumptions in the model. This additional P-source is interpreted as deriving from the use of an unresolved Dissolved Organic Phosphorus (DOP) pool by the implicit diazotrophs. Thus, in PISCES-v1, for every mole of  $N_2$ fixed by diazotrophy and instantaneously transferred into the ammonium pool, an additional 0.04 moles of phosphorus is added in the phosphate pool to represent the subsequent remineralization of the diazotrophy-derived organic matter with an N:P ratio of 46:1 (Figure 2a). In PISCES-v2, this parameterization has been changed and instead includes an unresolved source of inorganic phosphorus from labile DOP, independently of N-fixation. In strongly Plimited areas, diazotrophic cyanobacteria use DOP as a source of P, but this is the case also for other phytoplankton groups (Cotner et al, 1992; Paytan and McLaughlin, 2007). The parameterization thus mimics this source of P, which depends on simulated dissolved organic matter concentrations and is inhibited when dissolved inorganic P is not limiting phytoplankton growth (Figure 2b). In PISCES-v2, this P-source is therefore not dependent on the rate of Nfixation as it is in PISCES-v1".

Some specifics:

26 change "model" to "models"

**>> Changed accordingly.**

56 CESM2s is an official model name?

**>> CESM2s changed to "CESM2 and CESM2-WACCM".**

65 delete "atmospheric" (see also 94)

**>> Changed accordingly.**

97 "unconstrained" misspelled (see also 359, 502)

**>> Changed accordingly.**

160 change "advanced" to "modified"

**>> Changed accordingly.**

206 "phytoplankton-realised growth" do not hyphenate

**>> Changed accordingly.**

253 change "ocean physics" to "physical ocean"

**>> Changed accordingly.**

267, 269 change "independent from" to "independent of"

**>> Changed accordingly.**

270 " the reinforced relationship" Odd choice of words. How about: "confirms the intensified effect of diazotrophy on phytoplankton growth ..."?

**>> Text changed following the reviewer's suggestion.**

280 change "oceanic" to "ocean" (actually I think all 4 occurrences of "oceanic" could be changed to "ocean", but this one in particular)

**>> Changed accordingly.**

297 "The increase in N-fixation is dampened everywhere as compared to PISCES-v2, with even regions where N-fixation decreases in PISCES-v2fix" How about: "The increase in Nfixation is small everywhere compared to PISCES-v2; there are even regions where Nfixation decreases in PISCES-v2fix"?

**>> Text changed following the reviewer's suggestion.**

303 change "only slightly increases" to " increases only slightly"

**>> Changed accordingly.**

362 "better performance scores" I would not use this term unless the 'score' is defined somewhere.

**>> The score that the sentence is referring is based on the use of RMSE. This is now stated in the revised manuscript.**

467 change "under a similar high-emission scenario" to "under similar high-emission scenarios"

**>> Changed accordingly.**

468 "additional" misspelled

**>> Changed accordingly.**

471 "of the of the"

**>> Changed accordingly.**

479 change "participated to" to "participated in"

**>> Changed accordingly.**

481/496 Riche and Christian: 2017 or 2018? (see also 98)

**>> Reference to Riche and Christian has been corrected. The exact reference is:**

**Riche, OGJ and Christian, JR 2018 Ocean dinitrogen fixation and its potential effects on ocean primary production in Earth system model simulations of anthropogenic warming. Elem Sci Anth, 6: 16. DOI: https://doi.org/10.1525/elementa.277**

485-494 It might be a good idea to mention Pahlow et al and Inomura et al in this paragraph, as they were described on 423 as presenting "more mechanistic models of N fixation".

**>> References to Pahlow et al. and Inomura et al. now added to this paragraph.**

Figure 7 is too low resolution. When I expand it to be readable, the fonts are blurry.

**>> Figure 7 now provided at higher resolution.**
* * *
**Responses to Referee #2:**

This study by Bopp et al. investigates how changes to N2 fixation affects NPP in a suite of climate model simulations (IPSL) with PISCES biogeochemistry under historical and future warming scenarios. They simulate 5 different model versions, 4 of which apply different formulations for biogeochemistry, most notably N2 fixation, and 1 simulation with different ocean physical circulation and resolution. The simulation with higher resolution (IPSLCM6A) did not significantly affect N2 fixation compared to the simulation with similar biogeochemistry and lower resolution (PISCES-v2) so they mainly focused this study on the differences in N2 fixation parameterizations.

They implemented a variety of different N2 fixation parameterizations including phosphorus limitation, temperature-dependent growth, elemental stoichiometry, and underlying biogeochemistry (the latter not well described in the paper). Their model simulation with a strong increase in N2 fixation (PISCES-v2) caused a strong increase in NPP, whereas the other simulations with a slight increase or decrease in N2 fixation projected a decrease in NPP throughout the 21st century. They focus on a region in the western tropical North Pacific

to better understand the mechanisms responsible for the difference in these simulations. The one clear driver for driving the high N2 fixation rates in PISCES-v2 was the exponential temperature-dependence growth rate, whereas the

PISCES-quota simulations using the Breitbarth et al. 2007 bell-shape reduced diazotroph growth rate at high >26°C temperatures.

Overall I think this is a fine manuscript that highlights the importance of N2 fixation on NPP, which is often neglected or not analyzed in model simulations. Another aspect I liked was how the authors distinguished between NPP and phytoplankton biomass instead of only focusing on NPP like most studies.

**>> We thank Referee #2 for his positive comments on our manuscript.**

My one major issue with the manuscript is the insufficient description of which processes are causing N vs. P vs. Fe limitation and thus driving the N2 fixation response in the different biogeochemistry model simulations, but this can be addressed in revisions. This information is included in some of the references, but given the emphasis on N2 fixation in the paper I think some additional details should be included and discussed in this manuscript as well.

**>> We agree with Reviewer #2 that a more detailed description of the different PISCES versions used here was needed in our manuscript. We have expanded the description of the different model versions in the Methods section, which now reads:**

"In both PISCES versions, N-fixation is represented implicitly as a source of ammonium, i.e. without an explicit diazotroph plankton functional type (Figure 2). N-fixation is restricted to warm-waters (> 20°C) and increases exponentially with temperature following a Q-10 value of 1.9 as for all autotrophic processes in PISCES (Aumont and Bopp, 2006; Aumont et al., 2015). *N*-fixation is limited by the availability of light and iron, and favoured in low-nitrogen (NO3 and *NH*4) environments. PISCES-v1 and PISCES-v2 differ in their treatment of phosphorus limitation on N-fixation. which is absent in PISCES-v1 but combined with iron limitation in PISCES-v2. In PISCES-v1, due to the fixed stoichiometric ratios between carbon, nitrogen and phosphorus in all organic components it is assumed that N-fixation is accompanied by a release of inorganic phosphorus to account for the fact that diazotrophy-derived organic matter is much richer in N than the standard Redfield assumptions in the model. This additional P-source is interpreted as deriving from the use of an unresolved Dissolved Organic Phosphorus (DOP) pool by the implicit diazotrophs. Thus, in PISCES-v1, for every mole of  $N_2$ fixed by diazotrophy and instantaneously transferred into the ammonium pool, an additional 0.04 moles of phosphorus is added in the phosphate pool to represent the subsequent remineralization of the diazotrophy-derived organic matter with an N:P ratio of 46:1 (Figure 2a). In PISCES-v2, this parameterization has been changed and instead includes an unresolved source of inorganic phosphorus from labile DOP, independently of N-fixation. In strongly Plimited areas, diazotrophic cyanobacteria use DOP as a source of P, but this is the case also for other phytoplankton groups (Cotner et al, 1992; Paytan and McLaughlin, 2007). The parameterization thus mimics this source of P, which depends on simulated dissolved organic matter concentrations and is inhibited when dissolved inorganic P is not limiting phytoplankton growth (Figure 2b). In PISCES-v2, this P-source is therefore not dependent on the rate of Nfixation as it is in PISCES-v1".

**Major Comment**

The one critical issue I do not understand in this study is why N2 fixation in IPSL-CM5A-LR does not increase more similarly to IPSL-CM6A-LR. Since they have the same temperature

dependent growth rate, the authors state this is "due to the Ln term (limitation of excess inorganic nitrogen)" (line 339). However, all N2 fixation parameterization contain this Ln term (Figure 2). Why does Ln only decrease in IPSL-CM5A-LR and not any of the other simulations? My guess is that it is caused by some changes in the underlying biogeochemistry (e.g. iron limitation or denitrification), which is only mentioned in one brief sentence in the description section 2.2 (lines 143-145) and not further discussed. The processes that contribute to the decreasing Ln term in IPSL-CM5A-LR but not the other simulations should be described and discussed to understand these results presented here.

**>> Again, we agree with Reviewer #2 (Reviewer #1 has a similar comment). The method has been expanded (see above) and the section describing the mechanisms driving the differential responses of N-fixation to climate change has also been rewritten and made clearer. The new section reads:**

"To explain these contrasting trends, we focus on an area located in the northwestern tropical Pacific (130°E-160°E, 10°N-20°N), where the response of nitrogen fixation diverges strongly between IPSL-CM5A and IPSL-CM6A (Figure 5a) and exploit the comparison between the different offline versions of PISCES (PISCES-v1, PISCES-v2, PISCES-v2fix and PISCES-quota) that use an identical climate forcing. In this region, N-fixation increases by 32% in PISCES-v2 between 1986-2005 and 2080-2099 (of the same order of magnitude as in IPSL-CM6A, +45%, not shown), decreases by 46% in PISCES-v1, and decreases by 2% and 10% in PISCES-v2fix and PISCES-quota, respectively (Figure 5b). Concurrently, sea surface temperature in this region increases by nearly 4°C (between 1986-2005 and 2081-2100) to reach 31.5°C at the end of the 21st century. In PISCES-v1 and PISCES-v2, this increase leads to a boost in N-fixation by a factor of 2.1 (Figure 5d). In PISCES-v2fix and PISCES-quota, on the contrary, the increase in temperature reduces N-fixation by more than 30% (Figure 5d) due to the the bell-shaped temperature sensitivity function of the Breitbarth et al. (2007) parameterization.

In PISCES-v2, the limitation terms due to light, phosphate, iron and excess nitrate remain inoperative and N-fixation responds almost exclusively to temperature (Figure 5b). Indeed,  $NO_3$  concentrations remain close to zero (with N\* values, defined as  $NO_3 - 16*PO_4$ , remaining slightly negative), and the LN term has no influence, allowing N-fixation to increase in response to temperature. In PISCES-v1, on the other hand, the higher and increasing nitrate concentrations (Figure 5e), resulting in positive N\* values (Figure 5h), lead to a limitation and even to a decrease in N-fixation due to the LN term (limitation by excess inorganic nitrogen, Figure 5f) (see Figure 2b, equation (2)).

The differential response of N-fixation in PISCES-v1 and PISCES-v2 is thus related to how the parameterization of N-fixation simulates inorganic N and P inputs to surface waters (See Methods 2.2). In PISCES-v1, surface waters in the oligotrophic subtropical gyres are P-limited for phytoplankton (N\* being positive). Warming first increases N fixation, resulting in a continuous addition of N (through N-fixation) at the expense of P. Nitrogen fixation is hence progressively limited by the accumulation of inorganic nitrogen and thus decreases. In PISCES-v2, warming also increases N-fixation due to the same parameterization of the thermal sensitivity of N-fixation. But in this version, the sustained low addition of phosphate (which accounts for the implicit remineralization of DOP and is independent of N-fixation rates) prevents any shift to P limitation. N\* remains negative, inorganic N does not accumulate, and N-fixation continues to increase.

In PISCES-v2fix and PISCES-quota, NO3 concentrations remain close to zero as in PISCES-v1 (Figure 5e) and the excess nitrogen limitation term has no influence, remaining close to 1 (Figure 5f). But contrary to PISCES-v1, N-fixation decreases slightly (Figure 5b) due to warming and the use of the bell-shape temperature sensitivity function (Figure 5d).

Although the analysis presented here is limited to a small region of the western tropical Pacific (130E-160°E, 10°N-20°N), we show the same contrasting behavior between the different PISCES versions in other sub-tropical oligotrophic gyres (see Figure S1 showing the same analysis in a region centred around the HOT station, 175°E-205°E, 15°N-25°N).

In summary, the simulated N-fixation response is highly sensitive to (1) the parameterization used for the temperature sensitivity (PISCES-v1 and PISCES-v2 as compared to PISCES-v2fix and PISCES-quota), and (2) the respective evolution of NO3 and PO4 concentrations (PISCES-v1 as compared to PISCES-v2). To reiterate, it is the divergent responses of N-fixation in oligotrophic gyres (Figure 5a), where temperatures can exceed the optimal values in the Breitbarth et al. (2007)'s parameterization (Figure 2), and where NO3 accumulation can induce a decrease in the rate of N-fixation, that explain the differences between the different versions of PISCES'.

Minor Comments :

Lines 144-145: "... external sources of nutrients and the treatment of water-sediment interactions" Following my comment above, these processes can significantly impact N2 fixation rates. While I understand they are described in previous papers, the important processes controlling N2 fixation in these simulations presented here should be described and discussed.

>> We agree with the reviewer and have included more details on the external sources (rivers, atmospheric deposition) and sinks (sediment burial, denitrification) of nitrogen. The added paragraph reads:

"In both PISCES versions, the modelling of the nitrogen cycle relies on the explicit representation of nitrate and ammonium concentrations in seawater. It includes nitrification, which corresponds to the conversion of ammonium to nitrate and is assumed to be photo-inhibited and reduced in low-oxygen waters, as well as denitrification, when nitrate is used instead of oxygen for remineralization in suboxic waters (Aumont et al. 2015). External sources of nitrogen in the ocean include riverine input (using GLOBAL-NEWS2 data sets, Mayorga et al., 2010, for both PISCES versions), atmospheric deposition (using input4MIPs data, Heglin et al. 2016 in IPSL-CM6A-LR, and output of the INCA model, Aumont et al. 2008, for IPSL-CM5A-LR), and biological N-fixation (see below). External sinks of nitrogen include denitrification and organic matter burial in the sediment (see Aumont et al. 2006, and Aumont et al. 2015 for detailed descriptions)".

Line 155: "... phosphorus is added to the phosphate pool" I thought diazotrophs are responsible for consuming PO4 and DOP to very low levels e.g (Mather et al., 2008), not providing a source. These fluxes are generally small among the entire circulation-biogeochemical system so it is likely not significant, but I am concerned that this PO4 source could be supporting additional NPP in an unrealistic way.

>> The reviewer is right in assuming diazotrophs are consuming PO4 and DOP. But due to the constant-ratio used for all organic pools in most model versions (PISCES-v1, PISCES-v2, PISCES-v2fix), we account for an unresolved source of phosphate derived from the remineralization of an implicit labile dissolved organic phosphorus pool. In PISCES-quota, this additional source is not required as DOP is explicitly modeled. The very similar N-fixation changes with climate change simulated by PISCES-v2fix and PISCES-quota suggest that our (crude) parameterization is acting in a consistent manner.

Line 158: "... annual restoring of global mean PO4 concentration ..." I am surprised to see a PO4 restoring term in a prognostic climate model. I assume this is applied only during the spin-up and not historical/future projections? Does this occur at all depths?

>> The restoring of global mean PO4 concentration conserves the global phosphate pool – this is especially important for the long spin-up simulation of the coupled model so that the global P budget does not drift. This restoring term is applied everywhere (at all depths) and modifies phosphate concentration relatively, thus acting preferentially in the deep ocean where PO4 concentrations are higher. Overall, this term represents a restoring timescale of about 10,000 years.

The restoring term is retained for all simulations used here (historical/future projections included), but represents a very limited flux, which compensates for initial disequilibrium between P-input from rivers and atmospheric deposition (and implicit DOP remineralization), and P-loss through sediment burial.

Line 306: "The regional changes of NPP resemble those simulated in IPSL-CM5A" Is this also true for N2 fixation?

**>> Yes. This is true also for N2 fixation – the text will be modified accordingly.**

Lines 307-309: "... differences between PISCES-quota and two other offline simulations originate from the major developments in PISCES-quota such as variable C:N:P stoichiometry and the inclusion of a third phytoplankton ..."

This comes back to my major comment. It is interesting that newest (PISCES-quota) and the oldest formulation presented (PISCES-v1 in IPSL-CM5A-LR) give very similar projections of changes to NPP and N2 fixation (Figure 5a,b), despite their large differences in biogeochemical model structure/parameterizations, whereas the in-between version (PISCES-v2) shows a significant difference. I would be interested to know which changes mainly contributed to the large increase in N2 fixation and NPP and PISCES-v2 compared to v1. This statement specifically mentions stoichiometry and the additional phytoplankton type, but I thought it was pretty clear that the change in temperature-dependent N2 fixation growth rate was responsible for these decreases in N2 fixation and NPP in PISCES-quota compared to v2 as described in the following section so I am confused by this statement.

>> Section 3.3, describing the mechanisms explaining the divergent responses of Nfixation across our different model versions has been expanded and rewritten (see above). As described above, the respective role of DIN-inhibition (limiting N-fixation in PISCES-v1) and warming (limiting N-fixation in PISCES-v2fix and PISCES-quota) is now clearer (see above).

It is indeed interesting to note that our 2 "extreme" model versions, the least complex (PISCES-v1) and the most complex (PISCES-quota) display quite similar trends for N-fixation and subsequently NPP. If we trust the more advanced version (explicit variable stoichiometric ratios for C:N:P, no need for an implicit DOP remineralization, inclusion of a third phytoplankton functional group) more than the others, this would suggest that the simulated Nfixation / NPP response in PISCES-v1 (within IPSL-CM5) is "correct" but for the wrong reasons. It also demonstrates the importance of thermal performance curves, with that used in PISCES-v2fix and PISCES-quota, (an optimal temperature of 26°C being questionable, Fu et al. 2014).

Please describe how variable C:N:P stoichiometry and a third phytoplankton class causes higher N2 fixation because this is not clear. I think this emphasizes the need to better understand the key processes causing the N2 fixation differences in the different simulations.

>> Section 3.3, describing the mechanisms explaining the divergent responses of Nfixation across our different model versions has been expanded and rewritten (see above).

Mather, R. L., Reynolds, S. E., Wolff, G. A., Williams, R. G., Torres-Valdes, S., Woodward, E. M. S., Landolfi, A., Pan, X., Sanders, R., and Achterberg, E. P.: Phosphorus cycling in the North and South Atlantic Ocean subtropical gyres, Nature Geoscience, 1, 439-443, 10.1038/ngeo232, 2008.

**References cited:**

Cotner, J.B., Jr., and Wetzel, R.G. (1992). Uptake of dissolved inorganic and organic bphosphorus compounds by phytoplankton and bacterioplankton. Limnology and Oceanography 37, 232–243.

Fu, F.-X., Yu, E., Garcia, N. S., Gale, J., Luo, Y., Webb, E. A., and Hutchins, D. A. (2014) Differing responses of marine N2 fixers to warming and consequences for future diazotroph community structure, 72, 33–46.

Paytan, A., and McLaughlin, K. (2007). The Oceanic Phosphorus Cycle. Chem. Rev. 107, 563–576.

---

## Author Response (AR2)

Dear Editor,

We would like to thank you for the acceptance of our revised manuscript.

In the final version of our manuscript, we have:

- revised all figures so that they are all color-blind proof.

- added sections on Code/Data availability, Author contributions, and Competing Interests.

We would like to thank you for your time and understanding,

Best regards,

Laurent Bopp, on behalf of all co-authors